# Designing receptor agonists with enhanced pharmacokinetics by grafting macrocyclic peptides into fragment crystallizable regions

Katsuya Sakai [1,2] ✉, Nozomi Sugano-Nakamura[3], Emiko Mihara [3], Nichole Marcela Rojas-Chaverra [1], Sayako Watanabe[3], Hiroki Sato[1,4], Ryu Imamura[1,2], Dominic Chih-Cheng Voon [5,6], Itsuki Sakai[1], Chihiro Yamasaki[7], Chise Tateno[7], Mikihiro Shibata[2,8], Hiroaki Suga [9], Junichi Takagi [3] ✉ & Kunio Matsumoto [1,2,4] ✉

Short half-lives in circulation and poor transport across the blood–brain barrier limit the utility of cytokines and growth factors acting as receptor agonists. Here we show that surrogate receptor agonists with longer half-lives in circulation and enhanced transport rates across the blood–brain barrier can be generated by genetically inserting macrocyclic peptide pharmacophores into the structural loops of the fragment crystallizable (Fc) region of a human immunoglobulin. We used such 'lasso-grafting' approach, which preserves the expression levels of the Fc region and its affinity for the neonatal Fc receptor, to generate Fc-based protein scaffolds with macrocyclic peptides binding to the receptor tyrosine protein kinase Met. The Met agonists dimerized Met, inducing biological responses that were similar to those induced by its natural ligand. Moreover, lasso-grafting of the Fc region of the mouse anti-transferrin-receptor antibody with Met-binding macrocyclic peptides enhanced the accumulation of the resulting Met agonists in brain parenchyma in mice. Lasso-grafting may allow for designer protein therapeutics with enhanced stability and pharmacokinetics.

The clinical use of cytokines and growth factors as therapeutics has been approved by the US Food and Drug Administration[1,2], and their potential application in emerging areas, such as neurogenesis and brain repair[3,4], is a topic of intense research. However, their inherent structural properties render them challenging to engineer for improved physicochemical stability and pharmacokinetics, in particular, for

better half-life or blood–brain barrier (BBB) penetrance. Existing methods that control the pharmacokinetics of protein therapeutics include poly(ethylene glycol) conjugation and fusion with the fragment crystallizable (Fc) region of immunoglobulin[5–7] to extend their half-lives; and fusion with Fab of the anti-transferrin receptor (TfR) antibody to enhance their penetration across the BBB[8–10]. However, the

[1]Division of Tumor Dynamics and Regulation, Cancer Research Institute, Kanazawa University, Kanazawa, Japan. [2]WPI-Nano Life Science Institute (WPI-NanoLSI), Kanazawa University, Kanazawa, Japan. [3]Laboratory of Protein Synthesis and Expression, Institute for Protein Research, Osaka University, Suita, Japan. [4]Tumor Microenvironment Research Unit, Institute for Frontier Science Initiative, Kanazawa University, Kanazawa, Japan. [5]Inflammation and Epithelial Plasticity Unit, Cancer Research Institute, Kanazawa University, Kanazawa, Japan. [6]Cancer Model Research Innovative Unit, Institute for Frontier Science Initiative, Kanazawa University, Kanazawa, Japan. [7]Research and Development Department, PhoenixBio Co. Ltd, Higashihiroshima, Japan. [8]High-speed AFM for Biological Application Unit, Institute for Frontier Science Initiative, Kanazawa University, Kanazawa, Japan. [9]Department of Chemistry, Graduate School of Science, The University of Tokyo, Tokyo, Japan. ✉e-mail: k_sakai@staff.kanazawa-u.ac.jp; takagi@protein.osaka-u.ac.jp; kmatsu@staff.kanazawa-u.ac.jp

extent to which these methods can improve the pharmacokinetics of the proteins without adversely affecting their bioactivities is dependent on the properties of each protein[5–8]. To overcome the inherent structural limitations of cytokines and growth factors, there has been progress in the development of surrogate agonists that are structurally unrelated to native ligands[11,12]. Despite these recent advances, there remains a large unmet need for more robust and versatile methods to design protein therapeutics with desired physicochemical stability and pharmacokinetics.

Macrocyclic peptides have emerged as a promising novel class of drug candidates boasting a number of desirable features, such as antibody-like binding affinity and specificity[13,14], the ability to target unique chemical spaces[15–18] and efficient discovery through both rational/computational design and in vitro display[18–21]. In general, macrocyclic peptides display greater affinities for targets than linear peptides due to their constrained cyclic structures. An intriguing possibility in extending the applicability of these macrocyclic peptides is to 'engraft' them onto protein scaffolds to allow for functional combinations of peptide and protein[18,22–26]. However, grafting de novo identified peptides to protein loops has been challenging due to the potential misfolding of both the grafted peptide and the host protein[26].

The development of the RaPID (random non-standard peptides integrated discovery) system, which integrates messenger RNA display and genetic code reprogramming, has enabled the discovery of thioether-based cyclized macrocyclic peptides with exquisite binding specificity against target proteins[16–18]. In previous studies, we have shown that RaPID-derived pharmacophore sequences can be readily implanted onto surface-exposed loops of proteins and maintain both functions of the guest peptide and the host protein, which we termed 'lasso-grafting'[27–29]. The observed exceptional grafting compatibility is probably due to the intrinsic property of the pharmacophore motifs to self-fold into target-binding conformation similar to the parental macrocycle, even in the context of unrelated loop structure of the scaffold proteins[18,30].

In this study, we leverage the desirable scaffolding properties of the Fc fragment, its long half-life, versatility in combination with Fab[5,6,31,32] and ease of production to show the feasibility and application of lasso-grafting. Using the Fc as a scaffold, we generated Met receptor agonists that are characterized by markedly improved half-life and BBB penetrance.

## Results

### Met agonists generated by lasso-grafting Fc
Met is a receptor tyrosine kinase activated upon dimerization triggered by its ligand, hepatocyte growth factor (HGF). Met-HGF is centrally involved in morphogenesis during development, wound repair and organ homoeostasis by stimulating cell growth, survival and migration[33,34]. Recombinant HGF protein displays therapeutic efficacy in preclinical models[34,35] and in patients with specific diseases[36,37]. However, its short half-life and inability to be delivered beyond the BBB have long limited its potential in medical applications[38,39]. To address these shortcomings, we inserted by lasso-grafting the linearized pharmacophore sequence of aMD4 or aMD5, two macrocyclic peptides that bind to the ectodomain of Met[40] into each of the eight loops of Fc (Fig. 1a). These two series of aMD4- or aMD5-grafted Fc fragments (hereafter Fc(aMD4) and Fc(aMD5), respectively), each displaying two Met binders within 31–42 Å of each other, have the potential to proximate two Met receptors on the cell surface. The expression levels of Fc(aMD4) and Fc(aMD5) in Expi293F cells were similar to that of the control Fc, with the exception of Fc(aMD5)T3 (Extended Data Fig. 1), suggesting that lasso-grafting generally preserved the high expression of Fc.

The activation of cellular Met by purified Fc(aMD4) or Fc(aMD5) was highly dependent on the grafting site (Fig. 1b,c), where B3 was the optimal site for aMD4 (Fig. 1b). Our analyses show that Fc(aMD4)B3 activated Met with an half maximal effective concentration

($EC_{50}$) comparable to that of ungrafted aMD4-dimer peptide ($EC_{50}$: $4.5 \pm 0.2$ nM vs $5.1 \pm 0.6$ nM, respectively), and a slightly reduced maximum activation relative to HGF-induced phosphorylation ($74.8\% \pm 0.4\%$ vs $101.3\% \pm 0.1\%$, respectively). In contrast, grafting aMD4 into other loops resulted in lower Met activation relative to aMD4-dimer peptide. In the case of aMD5, B1 was the best grafting site, where Fc(aMD5)B1 activated Met with a 3.7-fold higher $EC_{50}$ compared with ungrafted aMD5-dimer peptide, and a slightly lower maximum activation ($EC_{50}$: $16.6 \pm 1.6$ nM vs $4.5 \pm 0.2$ nM, respectively; maximum activation: $76.2\% \pm 1.4\%$ vs $105.0\% \pm 4.2\%$, respectively; Fig. 1c). Grafting aMD5 into other loops either reduced or abolished Met activation. Notably, the agonistic activities of lasso-grafted Fc correlated well with their binding strengths to cell surface Met (Supplementary Fig. 1a) or Met ectodomain fragment ($Met_{ECD}$) (Extended Data Fig. 2 and Supplementary Fig. 1b), indicating that their efficacies are determined largely by their affinity to Met. The difference in the graft-site dependency of the two peptides is probably due to the steric effects imposed by the difference in their binding site on the Met ectodomain and their divergent spatial arrangements in the Fc structure (Supplementary Fig. 2).

To improve agonistic activity, we appended a Cys residue at each end of the aMD4 sequence to promote a tightly closed macrocyclic conformation via disulfide linkage (Fig. 2a and Supplementary Fig. 3). The disulfide-bonded derivatives aMD4 (hereafter prefixed with 'ds') at T1 and B3 showed greater maximum Met activation than the control (T1, $P = 0.039$; B3, $P < 0.0001$), while no effects were observed in the $EC_{50}$ (Supplementary Fig. 3f) and dissociation constant ($K_D$) values of $Met_{ECD}$ binding (Extended Data Fig. 2). In contrast, dsB2 showed no improvement in agonistic activity, maximum activation or $EC_{50}$ values (Supplementary Fig. 3f), suggesting that the reduced Met affinity after grafting into these two sites was not due to suboptimal conformation of the grafted peptide motif per se. This was further supported by the similar conformations of aMD4 or aMD5 when grafted at the B1 or B3 position (Supplementary Fig. 2).

The cellular responses induced by Fc(aMD4)B3 in EHMES-1 cells or in hepatocytes were highly comparable to those of HGF (Fig. 2 and Extended Data Fig. 3). Fc(aMD4)B3 at 1 nM induced phosphorylation of Met at Y1234/1235 and activated subsequent downstream signalling with phosphorylation of Akt and Erk1/2 at levels comparable to those induced by HGF (Fig. 2b and Supplementary Fig. 4). The levels of phosphorylation induced by Fc(aMD4)B3 and HGF at three different tyrosine residues of Met, as well as the induction kinetics of Met, Akt and Erk1/2 phosphorylation, were comparable (Extended Data Fig. 3). The selectivity of Fc(aMD4)B3 for Met was confirmed by a survey of tyrosine phosphorylation of 49 receptors (Fig. 2c and Supplementary Fig. 5). The changes in gene expression induced by Fc(aMD4)B3 or HGF were examined in primary human hepatocyte spheroid culture by RNA-seq (Fig. 2d–g). A scatterplot shows the correlation of gene expression changes induced by HGF and Fc(aMD4)B3 relative to untreated spheroids (Fig. 2e). Lastly, a heat map of differentially expressed genes (DEGs) relative to untreated spheroids indicates that both Fc(aMD4)B3 and HGF similarly altered the expression of more than 2,000 transcripts (Fig. 2f), with significant enrichment of gene ontology (GO) terms related to wound healing and liver functions (Fig. 2g).

### Mechanism of Met dimerization by Fc(aMD4)B3
As peptides generated de novo could differentially bind to the target receptors in a way completely different from that of the native ligands, we could theoretically generate agonists that do not compete with native ligands or are capable of activating mutated receptors that lack ligand binding. To investigate this potential, we first mapped the putative Fc(aMD4)B3 binding site (shown in red in Fig. 3a) using chimeric $Met_{ECD}$ fragments that variably fused human and mouse $Met_{ECD}$ (Extended Data Fig. 4), exploiting the specificity of aMD4 for human Met[40]. The putative binding site is mapped to the lower face of the

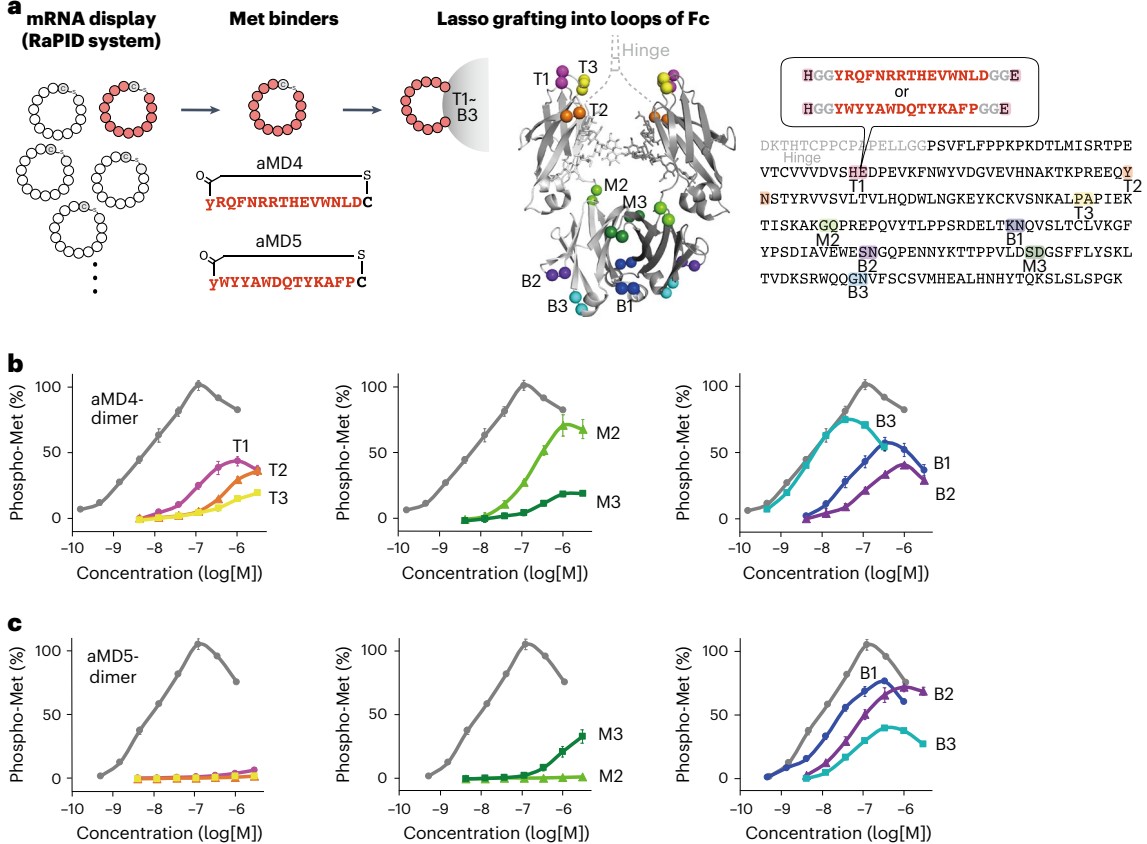

**Fig. 1 | Met agonists are generated by the insertion of Met-binding macrocyclic peptides into the structural loops of Fc protein. a,** Pharmacophore sequences of Met binders (aMD4 or aMD5; shown in red) were inserted into the loops (T1 to B3; coloured) of human IgG1 Fc protein (PDB ID: 1h3w). Amino acid sequences of Fc, grafted-peptides, and grafting sites are shown on the right. **b,c,** Cellular Met activation by aMD4-grafted Fc (Fc(aMD4)) (**b**) or aMD5-grafted Fc (Fc(aMD5)) (**c**). EHMES-1 cells were treated with Fc variants (coloured) or dimer peptides (grey). Cellular Met activation was quantified in situ with anti-phospho-Met (Tyr1234/1235) antibody. The results are shown as mean ± s.e.m. ($n$ = 6 independent experiments) in percentage of phospho-Met relative to maximum Met phosphorylation induced by 1.1 nM HGF. Left: T1, T2, and T3. Middle: M2 and M3. Right: B1, B2, and B3.

Sema domain of Met, which does not overlap with the HGF binding site[41], hence explaining why aMD4 peptide does not compete with HGF-induced Met activation[40].

We next examined the structure of the Met dimer induced by Fc(aMD4). Complexes between Met$_{ECD}$ and Fc(aMD4) were stabilized by crosslinking using bis(sulfosuccinimidyl)suberate (BS³) and analysed by sodium dodecyl sulfate-polyacrylamide gel electrophoresis (SDS–PAGE) (Fig. 3b and Extended Data Fig. 5a,b). The results indicated the formation of complexes between Met$_{ECD}$ and Fc(aMD4) in 1:1 or 2:1 stoichiometry. The relative efficiency of various Fc(aMD4) in forming a 2:1 signalling complex was in the order of B3 > B1 > B2, which correlated well with their ability to activate cellular Met. The BS³-crosslinked 2:1 complex of Met$_{ECD}$ and Fc(aMD4)B3 was purified by size exclusion chromatography (Extended Data Fig. 5c) and subjected to single-molecule examination by high-speed atomic force microscopy (HS-AFM)[42], with Fc and Met$_{ECD}$ as controls (Fig. 3c–f and Supplementary Videos 1–4). On HS-AFM, Met$_{ECD}$ showed a globular Sema domain and Ig-like, plexins, transcription factors (IPT) stalk domains, with the relative positioning of each IPT domain being flexible (Fig. 3d and Supplementary Video 2). Fc(aMD4)B3 bound to the lower face of the Sema domain and bridged two Met molecules while having the IPT stalk domains sprayed away (Fig. 3e and Supplementary Video 3) or attached (Fig. 3f and Supplementary Video 4). The flexibility of IPT domains probably allows for the proper association of the intracellular kinase domains of two Met molecules, this association being essential for Met activation[33,34]. These results show that while Fc(aMD4)B3 binds to Met on a site different from

that in HGF, its recruitment of two Met receptors in close proximity enables it to activate Met to the same extent as HGF.

## FcRn binding, half-life and in vivo activity

The long half-lives of Fc and antibodies are mainly maintained through their binding to the FcRn[43–46]. As such, we have selectively lasso-grafted into loop locations that are distal from the FcRn binding site (Fig. 4a). Accordingly, our analysis of binding kinetics between Fc(aMD4) and immobilized FcRn showed that lasso-grafting into most loops did not affect the affinity of Fc to FcRn, although Fc(aMD4)T2 and M2 exhibited slightly higher $K_D$ values than control Fc (Fig. 4b and Extended Data Fig. 6). This suggests that lasso-grafting on Fc preserves the affinity of Fc to FcRn.

We next evaluated the serum half-life of Fc(aMD4)B3 in comparison to those of control Fc and HGF in wild-type mice after a single intravenous (i.v.) administration at equimolar amount per kg (Fig. 4c). The serum concentration of HGF decreased to below 0.01 nM within 1 h, which was unable to activate Met. In contrast, Fc(aMD4)B3 showed an extended half-life comparable to that of control Fc (Fig. 4c). To accurately determine the pharmacokinetics of our human Fc-based variants, we measured the serum half-lives of Fc(aMD4)B3 and control Fc in mFcRn$^{-/-}$ hFcRn$^+$ mice[47] (Fig. 4d). The serum half-life of Fc(aMD4)B3 in these mice was 49.4 h and comparable to that of control Fc (46.6 h). The serum concentration of Fc(aMD4)B3 remained above 1 nM, the minimum concentration for activating Met (Fig. 2a,b), for up to 200 h after a single i.v. administration at 0.74 mg kg$^{-1}$ (Fig. 4d).

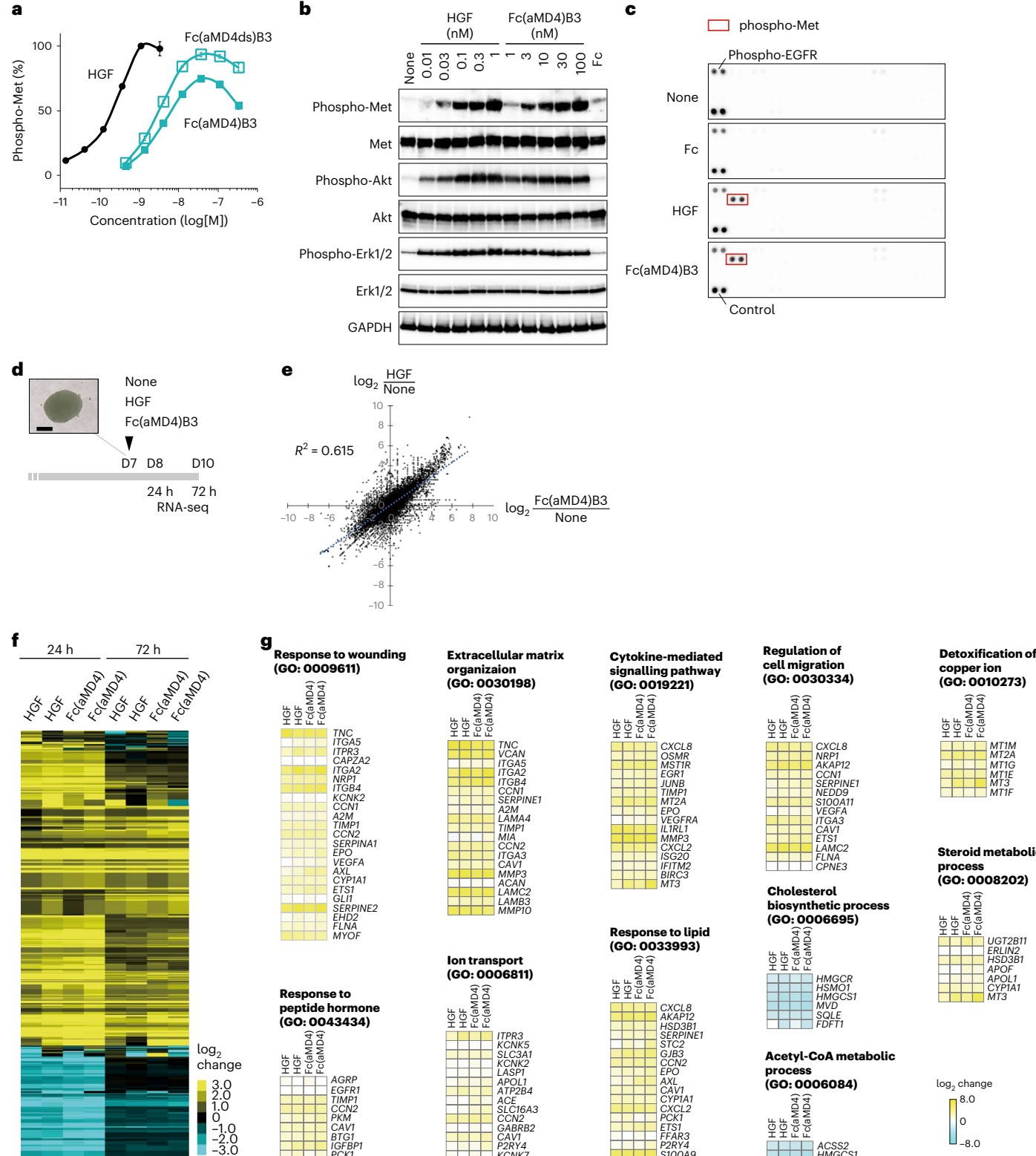

**Fig. 2 | Comparison of cell responses induced by aMD4-grafted Fc or HGF. a**, Cellular Met activation, as measured by phospho-Met (Tyr1234/1235), induced by Fc(aMD4)B3, Fc(aMD4ds)B3 and HGF in EHMES-1 cells. Results are shown as mean ± s.e.m. (*n* = 3 independent experiments). **b**–**g**, Hepatocyte response induced by Fc(aMD4)B3 is comparable to that induced by HGF. **b**, Cellular signalling in human hepatocytes treated with HGF, Fc(aMD4)B3 or control Fc (100 nM). The lysates were analysed by western blotting. **c**, Phospho-receptor tyrosine kinase arrays showing the selectivity of Fc(aMD4)B3 for Met. Human hepatocytes were stimulated with HGF, Fc(aMD4)B3 or Fc. **d**–**g**, Similar gene expression profiles in primary human hepatocyte spheroids induced by Fc(aMD4)B3 and HGF. **d**, Scheme of hepatocyte spheroid induction and stimulation by HGF or Fc(aMD4)B3. A representative image is shown. Scale bar, 200 μm. **e**, A scatterplot showing correlation between the gene expression changes induced by HGF and Fc(aMD4)B3 at 24 h. The blue dotted line indicates the regression line. **f**, Heat map of DEGs of HGF- or Fc(aMD4)B3-treated hepatocyte spheroids relative to untreated hepatocyte spheroids (*n* = 2 per group, false discovery rate (FDR) < 0.05, Benjamini–Hochberg, two-sided). **g**, Representative significantly enriched GO terms and their heat maps for DEGs of HGF- or Fc(aMD4)B3-treated hepatocyte spheroids relative to untreated controls (24 h, FDR < 0.05, Benjamini–Hochberg, two-sided).

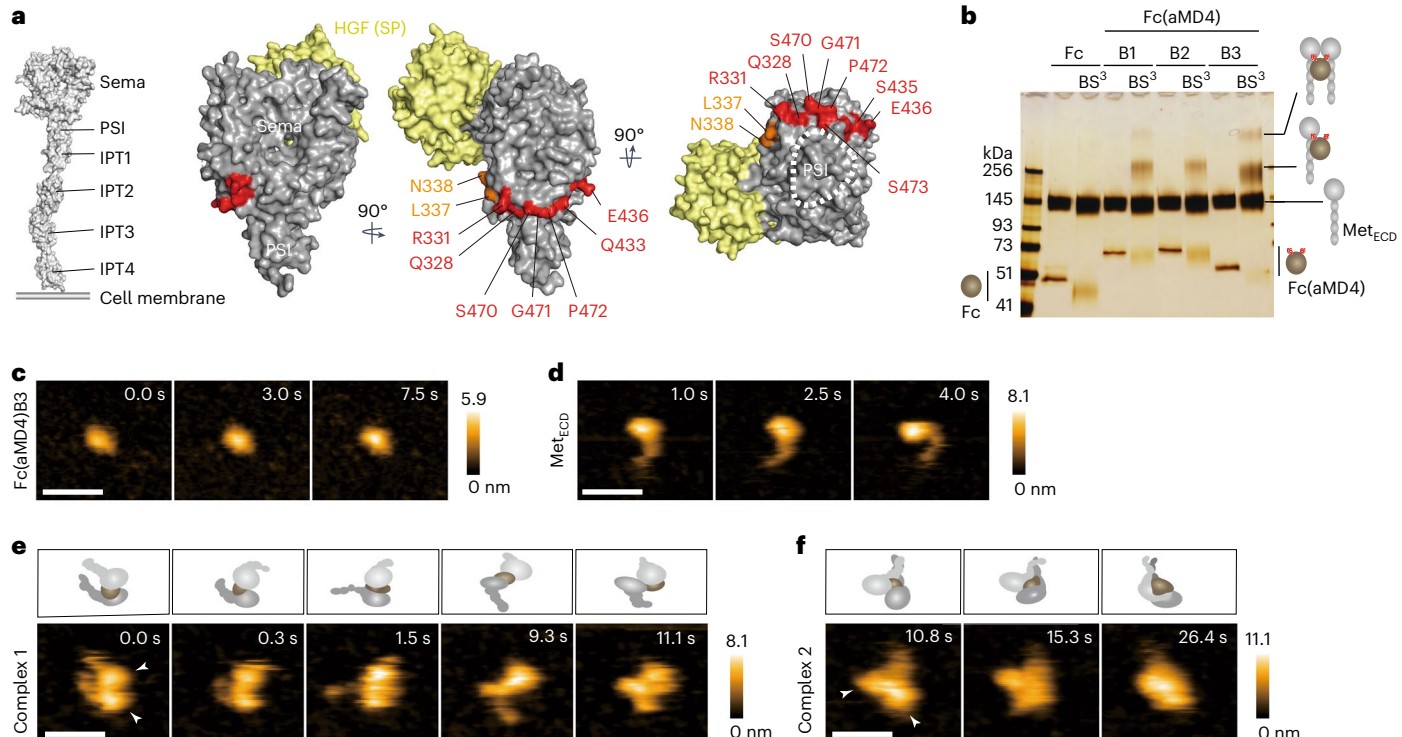

**Fig. 3 | Met dimerization is induced by aMD4-grafted Fc. a**, Essential (red) and auxiliary (orange) residues for the binding of Fc(aMD4)B3 in the Sema domain (PDB ID: 1shy). The SP domain of HGF is shown in yellow. Met ectodomain structure from predicted models (PDI ID: 1ux3 for Sema to IPT1, 2cew for IPT2 to IPT4). **b**, Met dimer formation by Fc(aMD4) in vitro. The complexes between Met_ECD and Fc(aMD4) or control Fc were crosslinked by BS³ and analysed by SDS–PAGE under non-reducing conditions and silver stained. **c**–**f**, Representative sequential HS-AFM images of Fc(aMD4)B3 (**c**), Met_ECD (**d**) and 2:1 complexes of Met dimer induced by Fc(aMD4)B3 (**e,f**). Schematic interpretations of the HS-AFM images are shown in **e** and **f** (top panels). Arrowheads indicate the Sema domain. Purification of the complex is shown in Extended Data Fig. 5c. At least five molecules or complexes were observed for each sample. The colours (from black to white) correspond to increasing heights of the molecules. Scale bars, 20 nm. These experiments were repeated independently twice, yielding comparable results.

The in vivo activity of Fc(aMD4)B3 was next examined using chimeric mice transplanted with human hepatocytes with approximately 80% of hepatocytes being humanized (PXB-mice)[48], as Fc(aMD4)B3 is incapable of activating murine Met[40]. We first confirmed the extended serum half-life of Fc(aMD4)B3 in PXB-mice after a single i.v. or subcutaneous (s.c.) injection at 5 mg kg⁻¹ (Supplementary Fig. 6). Subsequently, the proliferation and gene expression of human hepatocytes 46 h after a single s.c. injection of Fc(aMD4)B3 or PBS in PXB-mice (Fig. 4e–j) were analysed. Fc(aMD4)B3 significantly increased replicative DNA synthesis of hepatocytes compared with those of control mice treated with PBS (3.73% ± 0.31% vs 0.37% ± 0.07%, respectively, *P* < 0.0001) (Fig. 4f,g). A heat map of DEGs between Fc(aMD4)B3-treated and PBS-treated livers indicated that Fc(aMD4)B3 altered the expression of more than 3,200 transcripts (Fig. 4h). Significantly enriched GO terms among them were primarily involved in mitotic DNA replication/cell cycle and metabolic processes (Fig. 4i,j). Taken together, these results show that lasso-grafting generated a Met-activating Fc that is bioactive in vivo, with a markedly improved half-life that is comparable to that of unmodified Fc.

## Met agonists with BBB penetration

It has been reported that HGF-induced Met activation exerts beneficial neuroprotective effects in preclinical models of cerebral ischaemia, spinal cord injuries and neurological pathologies, such as Alzheimer's disease, amyotrophic lateral sclerosis and multiple sclerosis[35]. In line with earlier studies, we confirmed that Met is expressed in neurons in mouse brains (Supplementary Fig. 7). To generate Met agonists capable of crossing the BBB, we applied lasso-grafting of aMD4 on the Fc of the anti-mouse TfR (mTfR) antibody[49] in either single- or double-Fab configuration (sTfR(aMD4) and dTfR(aMD4), respectively) (Fig. 5a,b

and Supplementary Fig. 8). Lasso-grafting did not significantly alter the previously reported[9,10] affinity of these antibodies to mTfR, suggesting that Fab affinity is largely preserved (Fig. 5c). In contrast, both sTfR(aMD4) and dTfR(aMD4) showed reduced Met activation compared with Fc(aMD4), with 9.6- and 18.7-fold reduction in EC₅₀ values (20.2 ± 3.4 nM vs 39.2 ± 22.3 nM vs 2.1 ± 0.2 nM, respectively) (Fig. 5d). This was possibly due to the presence of large Fab arm(s) on sTfR(aMD4) and dTfR(aMD4), which may have interfered with their binding and/or dimerization of Met on the cell surface.

Lastly, we examined the BBB penetration of the three aMD4-grafted variants in wild-type mice after a single i.v. administration at therapeutically relevant doses of the proteins at equimolar amount per kg (Fig. 5e–j). Their concentrations in PBS-perfused brains or serum at 24 h post i.v. injection were determined by ELISA for human IgG Fc. Of the three variants, sTfR(aMD4) showed the highest percentage injected dose per gram brain (0.96% ± 0.07%), brain-to-serum ratio (8.3% ± 0.7%), as well as the highest brain concentration (10.7 ± 1.2 nM) (Fig. 5f–i), which is sufficient for Met activation (Fig. 5d). These results are in line with previous reports of low-affinity anti-TfR antibody accumulation in the brain[9,10,50]. In comparison, these values were significantly lower for dTfR(aMD4) and were nearly negligible in Fc(aMD4) (Fig. 5f–i). Concordant with the higher brain concentration determined by ELISA, sTfR(aMD4) showed prominent parenchymal staining co-localized with NeuN, GFAP and Iba1, indicating its distribution to neuron, astrocytes and microglia, respectively (Fig. 5j (middle right) and Supplementary Fig. 9). In comparison, dTfR(aMD4) showed less extensive neuronal staining and more endothelial staining (Fig. 5j (right)), in line with previous reports of high-affinity anti-TfR antibody accumulation in brain endothelial cells[9,10]. Lastly, Fc(aMD4) was undetectable in the

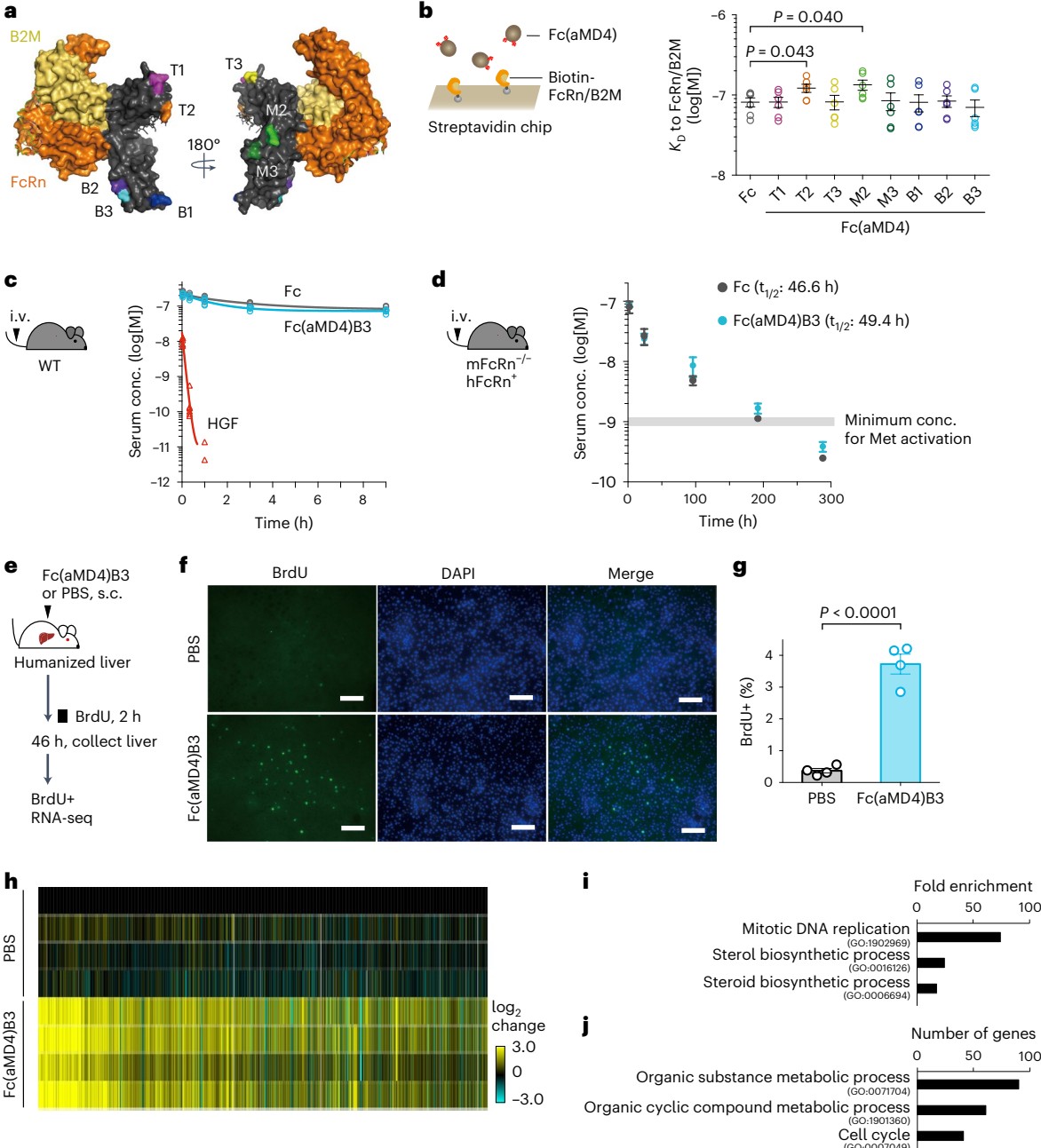

**Fig. 4 | Lasso-grafting Fc yields Met agonists with extended half-life and in vivo activity. a**, Peptide-inserted loops (T1 to B3) in Fc (grey) are designed away from the binding site to FcRn (orange) and β2-microglobulin (β2M, yellow) (PDB ID: 4N0U). **b**, $K_D$ values of Fc and Fc(aMD4) to FcRn/β2M determined by surface plasmon resonance. Results are shown as mean ± s.e.m. (duplicates of 3 concentrations of analytes; unpaired two-tailed *t*-test). **c,d**, Fc(aMD4)B3 showed a long serum half-life in mice. Wild-type C57BL/6 mice (**c**) or mFcRn$^{-/-}$ hFcRn$^+$ mice (**d**) were given a single injection of an equimolar amount per kg of Fc (0.7 mg kg$^{-1}$), Fc(aMD4)B3 (0.74 mg kg$^{-1}$) or HGF (1.0 mg kg$^{-1}$) via the tail vein, and serum concentrations were determined by ELISA. Results are shown as individual values (*n* = 6 mice per group) (**c**) or mean ± s.e.m. (**d**) (*n* = 10 mice

per group). For HGF, some samples were below the limit of detection. **e–j**, In vivo activity of Fc(aMD4)B3 in mice with humanized liver (PXB-mice). **e**, Schematic. **f,g**, Replicative DNA synthesis. **f**, Representative images showing BrdU staining in the liver of Fc(aMD4)B3- or PBS-treated mice. Scale bar, 100 μm. **g**, Percentage of BrdU+ cells in the liver. Results are presented as mean ± s.e.m. (*n* = 4 mice per group; unpaired two-tailed *t*-test). **h–j**, Gene expression profiles. **h**, Heat map of DEGs between Fc(aMD4)B3-treated and PBS-treated PXB-mice livers (*n* = 4 mice per group, FDR < 0.01, Benjamini–Hochberg, two-sided). **i,j**, Representative enriched GO terms in terms of fold enrichment (**i**) or number of genes (**j**) for DEGs of Fc(aMD4)B3-treated livers relative to PBS-treated control (FDR < 0.01, Benjamini–Hochberg, two-sided).

brain parenchyma (Fig. 5j (middle left)). Further evaluation of detailed pharmacokinetic profiles, including time course, will be needed for future therapeutic evaluation of sTfR(aMD4) in preclinical models. Taken together, these observations showed that we have generated BBB-penetrating Met agonists by lasso-grafting of Met-binding macrocyclic peptide on the Fc of anti-mTfR antibody.

## Discussion

The grafting of de novo identified peptides to improve their stability and bioavailability has been attempted on a variety of scaffolds[23–26]. These past studies revealed that successful grafting is highly dependent on the combination and compatibility between grafted peptides and scaffold proteins[25,26]. As such, the selection of scaffold protein

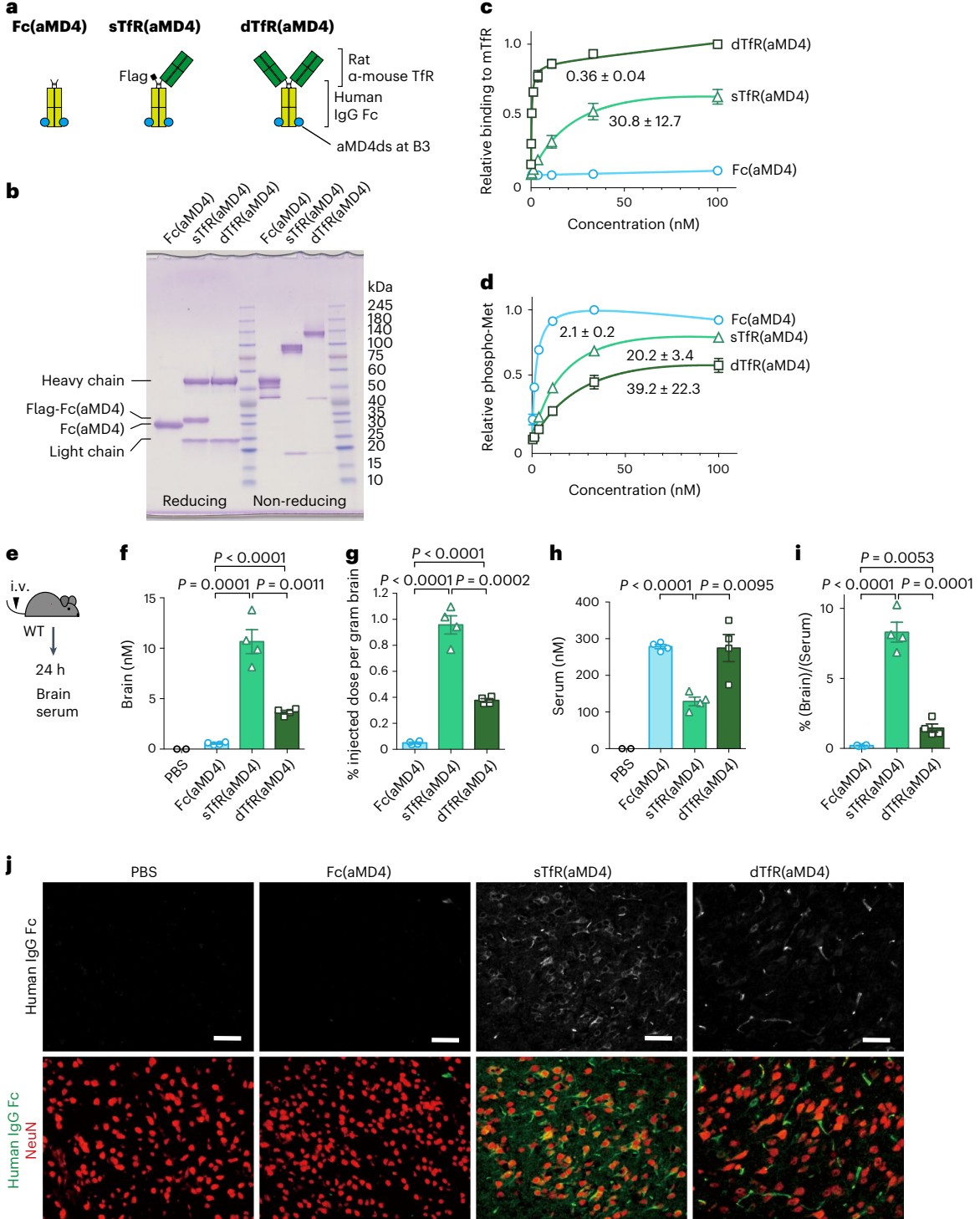

**Fig. 5 | Lasso-grafting Fc of anti-TfR antibodies yields Met agonists with BBB penetration. a**, Schematic representations of Fc(aMD4), dTfR(aMD4) and sTfR(aMD4). All variants were grafted with aMD4ds at the B3 loop of human IgG Fc. Rat anti-mouse transferrin receptor Fab was fused with Fc(aMD4). **b**, SDS–PAGE of purified variants. **c**, Lasso-grafting of aMD4 on anti-TfR antibodies preserved affinity to TfR. The apparent $K_D$ values (in nM) of Fc(aMD4), sTfR(aMD4) and dTfR(aMD4) to mouse TfR (mTfR) were determined by ELISA and are presented as mean ± s.d. ($n = 3$ independent experiments). **d**, Cellular Met activation by aMD4-grafted anti-TfR antibodies. EHMES-1 cells were treated with Fc(aMD4), sTfR(aMD4) or dTfR(aMD4). Cellular Met activation was quantified in situ with anti-phospho-Met (Tyr1234/1235) antibody and presented as mean ± s.d. ($n = 3$ independent experiments). $EC_{50}$ values (in nM) are indicated.

**e**–**j**, BBB penetration of aMD4-grafted anti-TfR antibodies. Schematic (**e**), brain concentration (**f**), percentage injected dose per gram of brain (**g**), serum concentration (**h**) and brain-to-serum ratio (**i**) at 24 h after a single injection of Fc(aMD4), sTfR(aMD4) or dTfR(aMD4) at equimolar amount per kg (12, 20 and 26.5 mg kg⁻¹, respectively) via the tail vein. Results are shown as mean ± s.e.m. ($n = 4$ mice per group; unpaired two-tailed $t$-test). **j**, Representative images of immunohistochemical staining of brain sections from mice at 24 h after a single injection of Fc(aMD4), sTfR(aMD4) or dTfR(aMD4) via the tail vein. Note the broad staining for sTfR(aMD4) in the brain parenchyma and localization around NeuN-positive neuronal cell bodies. Staining for dTfR(aMD4) was more prominent in endothelial cells than in neuronal cell bodies. Scale bars, 50 μm.

is a key consideration, as it determines the success of peptide grafting while conferring desirable properties such as metabolic stability and cell permeability to the engineered products[24–26]. To cater for this broad spectrum of desirable features, disulfide-stabilized peptides and miniproteins have been engineered to act as specialized scaffolds for peptide grafting[23–26]. However, these specialized scaffolds are small non-human proteins with no particular bioactivities of their own. Although natural human proteins are desirable scaffolds for peptide grafting, there has been very limited attempts at grafting de novo identified peptides onto natural human scaffolds so far. A major reason behind this is that grafting synthetic de novo identified peptides into an independently folded domain of a natural scaffold would often result in the misfolding and inactivation of both entities. It is therefore remarkable that our data thus far indicate that the functional grafting of de novo identified macrocyclic peptides into the protein loops of natural scaffolds is far more feasible than previously expected[27–29].

As a protein scaffold, the Fc is highly desirable because of its high expression, ease of manufacture, long half-life, and Fab compatibility[5,6,31,32]. In this study, we report that lasso grafting Fc with macrocyclic peptides yielded Met receptor agonists with extended half-life and BBB penetration. Our application of lasso-grafting on Fc has revealed three key strengths to this approach: Firstly, lasso-grafting preserved the overall biochemical characteristics of Fc in terms of expression level, FcRn affinity, and Fab compatibility. Secondly, lasso-grafted macrocyclic peptide largely preserved parental macrocyclic conformation even without disulfide linkage. Thirdly, there is a wide choice of suitable insertion positions in Fc. Our study further showed that the target affinity of grafted peptides is site-dependent. For example, the T-loops were less effective in our study, probably due to the presence of a N-terminus hinge region that could interfere with the binding of grafted peptides to the target protein, Met. Yet, these loops acted as suitable grafting positions for other receptor binders[27,29]. Notwithstanding this, the robust graft compatibility between macrocyclic peptides and Fc, in combination with a wide choice of insertion positions, renders lasso-grafting a particularly effective approach of Fc engineering.

An important implication of our findings is that lasso-grafting protein scaffolds with specific properties can serve as a framework for engineering designer receptor agonists with the desired properties based on the choice of protein scaffolds. As a demonstration of this, we generated BBB-permeable agonists by lasso grafting an anti-TfR antibody, thereby leveraging an established BBB-permeation technique[8–10]. At present, there are several efforts in developing anti-TfR Fab fusion antibodies with blocking properties, but only few that attempt to develop agonist antibodies[51]. Lasso-grafting could be used to design agonist BBB-permeating antibodies, in addition to antagonistic ones. Furthermore, one major drawback of current BBB-permeating antibodies is that, owing to their low brain-to-serum ratio and long half-life in circulation, systemic concentrations often exceed the desired range for extended periods. This could cause unwanted neo-immunogenicity or side effects, especially if the targeted molecule is functionally important in other tissues. With lasso-grafting, this could be avoided by grafting functional peptides directly into scaffold proteins with a short half-life, such as the anti-TfR Fab.

Contrary to existing methods that improve the pharmacokinetics of protein therapeutics by modifying the protein, lasso-grafting endows novel functions derived from small peptides to well-understood protein scaffolds[27–29], hence avoiding potential pitfalls such as reduced biological activity, unfavourable protein characteristics for manufacturing or neo-immunogenicity arising from protein fusion or from poly(ethylene glycol)[5–7]. Although Fc serves as an excellent scaffold because of its immunosuppressive property[52], the immunogenicity, safety and efficacy of lasso-grafted proteins await future evaluation as part of drug development. In the same vein, HGF has shown therapeutic efficacies in preclinical models of various diseases such as non-alcoholic steatohepatitis[53,54] or neurological pathologies[35,37]. The

therapeutic advantages of the lasso-grafted Met agonists generated in this study should likewise be further evaluated in disease models.

In view of the rapid advances in potent bioactive and graft-compatible macrocyclic peptides, lasso-grafting holds great promise for the generation of designer protein therapeutics with the desired physicochemical stability and pharmacokinetics, as well as for engineering multifunction proteins and antibodies.

## Methods

### Protein design, construction and purification

To design peptide-grafted Fc protein, Fc protein was structurally inspected to find an appropriate insertion point where the approximate distance between the two anchor point residues located within an exposed loop was less than 7 Å. For the construction of control or Met-binding macrocyclic peptide (aMD4 or aMD5)-grafted Fc, human IgG1 Fc (residues 104–330, Uni-Prot P01857) was used without any tags. The internal sequence of aMD4 or aMD5 appended with or without Cys and two spacer residues of Gly at both ends were inserted into these identified sites (T1 to B3). For the construction of anti-mouse TfR antibodies with aMD4ds insertion at the B3 site, the Fab fragment of a rat anti-mouse TfR monoclonal antibody (clone 8D3)[49] was used. The VH to CH1 fragment of 8D3 was fused to the N terminus of Fc(aMD4) to yield heavy chain (HC) of dTfR(aMD4). The VL to CL fragment of 8D3 was used to yield light chain (LC). For the construction of sTfR(aMD4), a flag tag was added at the N terminus of Fc(aMD4). The amino acid sequences used for s/dTfR(aMD4) can be found in Supplementary Fig. 8. All expression constructs were made using pcDNA3.1-based backbone with appropriate signal peptide and the coding region of all expression constructs was verified by DNA sequencing. Protein expressions were performed using the Expi293 expression system (Thermo Fisher). Secreted proteins were purified on a HiTrap Protein A HP column (Cytiva) using elution with 0.1 M citrate acid-sodium citrate buffer (pH 3.5), followed by neutralization, dialysis against phosphate-buffered saline (PBS) and size exclusion chromatography on a Superdex 200 Increase 10/300GL column (Cytiva) equilibrated with PBS. SDS–PAGE analysis and purity of purified Fc variants used in the assay can be found in Supplementary Fig. 10. For the expression and purification of sTfR(aMD4), three DNAs (TfR(aMD4) HC, TfR(aMD4) LC and Fc(aMD4)) were used to co-transfect Expi293F cells, yielding the mixture of sTfR(aMD4), dTfR(aMD4) and Fc(aMD4). The sTfR(aMD4) was purified from the latter two components by sequential anti-Flag affinity purification and size exclusion chromatography on a Superdex 200 Increase 10/300GL column equilibrated with PBS.

### Quantification of phosphorylation level of Met, Akt and Erk1/2 in situ

EHMES-1 human mesothelioma cells were seeded at 8,000 cells per well in a 96-well black µClear-plate (Greiner Bio-One) and cultured in RPMI1640 medium supplemented with 10% fetal bovine serum (FBS) for 24 h. Cells were stimulated with each protein in RPMI1640 medium supplemented with 10% FBS for 10 min or indicated time, washed with ice-cold PBS, fixed with 4% paraformaldehyde in PBS for 30 min, washed with PBS and blocked with 5% goat serum and 0.02% Triton X-100 in PBS for 30 min. Phosphorylation of Met, Akt or Erk1/2 was detected using an anti-phospho-Met (Tyr1234/1235) antibody (1:1,000 dilution, D26, Cell Signaling Technologies), phosphorylated Akt (S473) (1:1,000 dilution, D9E, Cell Signaling Technology) or phosphorylated Erk1/2 (T202/Y204) (1:1,000 dilution, D13.14.4E, Cell Signaling Technology), and a horseradish peroxidase (HRP)-conjugated anti-rabbit goat antibody (1:1,000 dilution, Dako). Then, chemiluminescence developed with ImmunoStar LD reagent (Wako) was measured using an ARVO MX plate reader (Perkin Elmer).

### Flow cytometry

The binding of peptide-grafted Fc proteins to Met-knockout CHO cells and Met-reconstituted CHO cells[27] was detected using flow cytometry.

Cells were detached from dishes by brief treatment with 0.25% trypsin and 1 mM ethylenediaminetetraacetic acid, plated at 200,000 cells per well and incubated with peptide-grafted Fc proteins diluted at 10 mg ml⁻¹ in 100 ml Ham's F-12 medium containing 5% FBS on ice for 1.5 h. After washing twice with ice-cold PBS, cells were incubated with Alexa Fluor 488-labelled goat anti-human IgG (1:400 dilution in 100 μl Ham's F-12 medium containing 5% FBS, Thermo Fisher, A11013) on ice for 30 min, then analysed on an EC800 system (Sony). Gate on forward scatter vs side scatter was set to include all cell populations but excluding debris and dead cells as shown in Supplementary Fig. 11. The data were analysed with FlowJo software (Tomy Digital Biology).

### Surface plasmon resonance analysis

The ectodomain fragment of human Met (Met$_{ECD}$, 1–931) was C-terminally fused with a biotin acceptor sequence (SSLRQILDSQK-MEWRSNAGG) and co-expressed with a biotin ligase (BirA) in Expi293F cells to achieve BirA-mediated biosynthetic biotinylation and immobilized onto a Series S Biotin CAPture chip (Cytiva) at a surface density of ~210 resonance units (RU). The binding of Fc and peptide-grafted Fcs was evaluated by injecting the analyte solutions serially diluted using the running buffer (20 mM HEPES-NaOH (pH 7.5), 150 mM NaCl, 0.05% Tween 20). The runs were conducted in single cycle kinetics mode employing the following parameters: flow rate of 30 μl min⁻¹, contact time of 120 s and dissociation time of 300 s. After each run, the surface was regenerated by injecting the regeneration buffer (6 M guanidine-HCl, 250 mM NaOH) until the response returned to the original baseline level. The binding curves were obtained by subtracting the RU of the reference cell (un-immobilized) from that of the measurement cell (immobilized with Met$_{ECD}$) and used to derive kinetic binding values. Data were obtained using a Biacore T200 instrument (Cytiva) at 25 °C, and the results were analysed using Biacore T200 evaluation software v3.2 (Cytiva). Biotinylated human FcRn/β2M (ACROBiosystems) was captured on the surface of streptavidin-coated chips (Cytiva) at 180-220 RU. Analytes were diluted in running buffer (50 mM phosphate (pH 6.0), 150 mM NaCl, 0.01% Tween 20) and injected at 30 μl min⁻¹ for 1 min, followed by dissociation for 0.7 min. The binding curves were obtained by subtracting the RU of the reference cell (un-immobilized) from that of the measurement cell (immobilized with FcRn/2M) and following axis-zeroing, sensorgrams were fit locally to a 1:1 Langmuir binding model. Data were obtained using a Biacore 3000 instrument (Cytiva) at 25 °C, and the results were analysed using BIAevaluation software v4.1 (Cytiva).

### Predicted structures of the CH3 domain of Fc with aMD4 or aMD5 insertion

Structures of the CH3 domain of Fc with aMD4 or aMD5 inserted in the B1 or B3 loop were predicted using ColabFold and AlphaFold2 in MMseqs2[55].

### Trypsin digestion and liquid chromatography–mass spectrometry (LC–MS/MS)

Trypsin digestion and LC–MS/MS analysis were performed by KANEKA TECHNO RESEARCH. One hundred μg of Fc(aMD4ds)T1 or Fc(aMD4ds)B3 in PBS were treated with 100 mM ammonium bicarbonate (Sigma-Aldrich) containing 8 M urea (Fujifilm Wako Pure Chemicals). After desalting using an Amicon Ultra device (MWCO: 10 K, Millipore), 10 μg of each protein was digested with trypsin (Promega) for 12 h at 37 °C under non-reducing conditions. The reaction was stopped by the addition of formic acid. The samples were injected into a Nexera X2 UHPLC system (Shimadzu corporation) connected to a SCIEX TripleTOF 6600 system (AB SCIEX) equipped with an ESI ionization source. Peptides were chromatographically separated on a C18 reverse-phase column with linear-gradient of mobile phase A (water containing 0.1% formic acid) and mobile phase B (acetonitrile (Fujifilm Wako Pure Chemicals) containing 0.1% formic acid). MS and MS/MS data were collected using IDA, positive ion mode. The LC–MS raw data were processed using SCIEX BioPharmaView software. The following criteria were used to identify peptides: mass accuracy for the matched precursor must be within 5 ppm of mass error, and peptide matching score was set to at least 3 for peptide identification.

### Crosslink and isolation of Fc(aMD4)/Met$_{ECD}$ complex

Met$_{ECD}$ (60 nM) and Fc(aMD4)B1/B2/B3 (20 nM) were mixed in 100 μl PBS with 0.01% Triton X-100 for 1 h. The samples were divided into two and treated with 1 mM BS³ (Thermo Fisher) or left untreated for 1 h, quenched by 50 mM Tris, subjected to SDS–PAGE (7.5% or 3–10% gel) and silver stained. For the HS-AFM observation, Met$_{ECD}$ (2.6 μM) and Fc(aMD4)B3 (1.3 μM) were mixed in 100 μl PBS with 0.01% Triton X-100 for 1 h, treated with 1 mM BS³ for 1 h, quenched by 50 mM Tris and analysed by size exclusion chromatography on a Superose 6 Increase 10/300GL column (Cytiva) equilibrated with PBS. The separated fractions were analysed by SDS–PAGE and stained with Coomassie brilliant blue. A single fraction was subjected to HS-AFM analysis.

### HS-AFM observations

HS-AFM measurements were operated in tapping mode. Cantilever deflection was detected with an optical beam deflection detector using an infrared (IR) laser (0.7 mW, 780 nm). The IR laser beam was focused onto the backside of a cantilever covered with gold film (Olympus, BL-AC10DS-A2) through a ×60 objective lens (CFI S Plan Fluor ELWD ×60; Nikon). Reflection of the IR laser from the cantilever was detected using a two-segmented PIN photodiode (MPR-1; Graviton). The free-oscillation amplitude was approximately 1 nm and the set-point amplitude was approximately 90% of the free amplitude for feedback control of HS-AFM. For the AFM probe, an amorphous carbon tip with a length of approximately 500 nm grown by electron beam deposition was used. For the AFM substrate, a mica surface treated with 0.01% 3-aminopropyl-triethoxysilane (Shin-Etsu Silicone) was used. All HS-AFM observations were performed in a buffer solution containing 50 mM Tris-HCl (pH 7.4) and 100 mM NaCl at room temperature.

### Hepatocytes monolayer culture

Monolayers of primary human hepatocytes derived from chimeric mice with humanized liver were obtained from PhoenixBio and grown in collagen-coated 12-well plates. Hepatocytes were cultured in dHCGM medium consisting of Dulbecco's modified Eagle's medium (DMEM) with 10% FBS, 20 mM HEPES, 44 mM NaHCO₃, antibiotics (100 U ml⁻¹ penicillin G and 100 μg ml⁻¹ streptomycin), 15 μg ml⁻¹ L-proline, 0.25 μg ml⁻¹ insulin, 50 nM dexamethasone, 5 ng ml⁻¹ epidermal growth factor, 0.1 mM L-ascorbic acid, and 2% dimethyl sulfoxide (all supplements from PhoenixBio).

### Hepatocyte spheroids

Cryopreserved human hepatocytes were obtained from Gibco. Cells were seeded into ultra-low-attachment 96-well plates (Nunclon Sphera, Thermo Fisher) at 1,500 viable cells per well and subsequently centrifuged at 200 × g for 2 min. Cells were seeded in 100 μl Williams E medium supplemented with 2 mM GlutaMAX, 100 U ml⁻¹ penicillin, 100 μg ml⁻¹ streptomycin, 4 μg ml⁻¹ insulin, 1 μM dexamethasone, 15 mM HEPES (pH 7.4) and 10% FBS (all supplements from Oriental Yeast). From five days after seeding, when the spheroids were sufficiently compact, 50% of the medium was exchanged every 2 d with serum-free Williams E medium supplemented with 2 mM GlutaMAX, 100 U ml⁻¹ penicillin, 100 μg ml⁻¹ streptomycin, 6.25 μg ml⁻¹ insulin, 6.25 μg ml⁻¹ transferrin, 6.25 μg ml⁻¹ selenous acid, 1.25 mg ml⁻¹ BSA, 5.35 μg ml⁻¹ linoleic acid, 100 nM dexamethasone and 15 mM HEPES (pH 7.4) (all supplements from Oriental Yeast). Spheroids were treated in a serum-free medium with 0.3 nM HGF or 30 nM Fc(aMD4)B3 on days 7 and 9. Forty-eight spheroids were collected for each group and subjected to RNA isolation at days 8 or 10.

## Western blotting and phospho-RTK array

Monolayer hepatocytes were serum starved in DMEM with 0.2% BSA for 3 h, stimulated with HGF (0.01-1 nM), Fc(aMD4)B3 (1-100 nM), Fc (100 nM) or unstimulated (None) in DMEM with 10% serum for 10 min. EHMES-1 cells were serum starved in RPMI1640 with 0.2% BSA for 3 h, stimulated with HGF (0.04-1.1 nM), Fc(aMD4)B3 (1.3-33 nM) or unstimulated in RPMI1640 with 0.2% BSA for 10 min. These cells were washed with ice-cold PBS, and lysed in 400 µl of lysis buffer (50 mM Tris, 150 mM NaCl, 1% Triton X-100, 1% NP-40, 10% glycerol, 1 mM phenylmethylsulfonyl fluoride, 1 mM $Na_3VO_4$, 1 mM NaF, 1 mM ethylenediaminetetraacetic acid and protease inhibitors). The lysates were passed through a 27G needle and 0.45 µm filter, and centrifuged. Protein concentrations of lysates were measured by bicinchoninic acid assay (Thermo Fisher). The supernatants were subjected to western blotting for phosphorylated Erk1/2 (T202/Y204) (1:1,000 dilution, D13.14.4E, Cell Signaling Technology), Erk1/2 (1:1,000 dilution, 137F5, Cell Signaling Technology), phosphorylated Akt (S473) (1:1,000 dilution, D9E, Cell Signaling Technology), Akt (1:1,000 dilution, 11E7, Cell Signaling Technology) or GAPDH (1:1,000 dilution, 14C10, Cell Signaling Technology). The supernatants were subjected to immunoprecipitation with an anti-Met antibody (5 µg for 600 µl lysates, D-4, Santa Cruz Biotechnology) and western blotting for Met (1:1,000 dilution, D1C2, Cell Signaling Technology), phosphorylated Met (Y1234/Y1235) (1:1,000 dilution, D26, Cell Signaling Technology), phosphorylated Met (Y1003) (1:1,000 dilution, 13D11, Cell Signaling Technology) or phosphorylated Met (Y1349) (1:1,000 dilution, 130H2, Cell Signaling Technology). The blottings were followed by HRP-conjugated secondary antibody (Dako) and measurement of chemiluminescence developed with ImmunoStar LD reagent (Wako), using the image reader FUSION-SOLO.6S. EDGE (VIRVER). For the phospho-RTK array, hepatocytes were serum starved in DMEM with 0.2% BSA for 3 h, stimulated with HGF (1 nM), Fc(aMD4)B3 (100 nM), Fc (100 nM) diluted in DMEM with 10% serum for 10 min, washed with ice-cold PBS, lysed in 400 µl of lysis buffer 17 (R&D Systems) with protease inhibitor cocktail (Nacalai Tesque), passaged through a 0.45 µm filter and centrifuged. The lysates (1 mg) were analysed using the Human Phospho-RTK Array kit (R&D Systems).

## RNA-seq

Total RNA of hepatocyte spheroids or RNAlater (Thermo Fisher)-treated livers from PXB-mice was prepared using the QIAzol lysis reagent (Qiagen) and shipped to the Bioengineering Lab (Kanagawa, Japan) for RNA quality control and RNA-seq on a DNBSEQ-G400 sequencer (MGI Tech). Complementary DNA libraries for RNA-seq were prepared using MGIEasy RNA directional library prep set (MGI Tech) and evaluated using AATI fragment analyzer (Advanced Analytical Technologies). The cDNA libraries were circularized using MGIEasy circularization kit (MGI Tech), prepared as DNA nano balls by DNBSEQ-G400RS high-throughput sequencing kit (MGI Tech), and followed by massively parallel sequencing (2×100 bp) on a DNBSEQ-G400 sequencer (MGI Tech). After excluding adaptor sequences using cutadapt (ver. 1.9.1) and sequences with low-quality score (<20) or less than 40 nucleotides using sickle (ver 1.33), reads were aligned to the reference human genome (GRCh38.p13) using the hisat2 software (v2.2.0). The alignment data were counted using featureCounts (ver. 2.0.0). Transcript abundances were measured in reads per kilobase of exon per million mapped reads (RPKM) and transcripts per million (TPM). DEG analysis was performed using DEGES in TCC (v1.18.0) and DESeq (v1.30.0) (FDR < 0.05 or <0.01, Benjamini−Hochberg, two-sided). All DEGs identified were subjected to cluster analysis using Gene Cluster 3.0. Significantly enriched GO terms were identified using GO Enrichment Analysis by PANTHER.

## Half-life in serum

Half-life in serum up to 9 h was determined in wild-type C57BL/6 mice (female, 9 weeks, 18−21 g, obtained from Japan SLC) given a single tail-vein injection at 0.7 mg kg$^{-1}$ for Fc, 0.74 mg kg$^{-1}$ for Fc(aMD4) B3 and 1.0 mg kg$^{-1}$ for human recombinant HGF (Kringle Pharma), diluted in 100 µl PBS. Half-life in serum up to 12 d was determined in mFcRn$^{-/-}$ hFcRn Tg 276 heterozygote mice (female, 9−12 weeks, 18−24 g) produced in-house by mating mFcRn$^{-/-}$ mice and hFcRn Tg 276 homozygote mice obtained from Jackson Laboratory. Mice were given a single tail-vein injection at 0.7 mg kg$^{-1}$ for Fc or 0.74 mg kg$^{-1}$ for Fc(aMD4)B3 diluted in 100 µl PBS. Half-life in serum up to 12 d was also determined in chimeric mice with humanized liver (PXB-mice, PhoenixBio, male, >12 weeks, 19−24 g) given a single tail-vein injection or subcutaneous injection at 5.0 mg kg$^{-1}$ for Fc(aMD4)B3, diluted in 100 µl PBS. Blood was drawn from the submandibular vein using a Goldenrod animal lancet (Bio Research Center) or from the orbital plexus at each time point, processed to serum and stored at −80 °C until analysis. The serum concentrations of Fc and Fc(aMD4)B3 were determined using a human immunoglobulin recognition immunoassay (human IgG ELISA quantitation set, Bethyl Laboratories). The serum concentrations of HGF were determined using an in-house developed ELISA. All animal experimental procedures were conducted in accordance with the guidelines provided by the Proper Conduct of Animal Experiments (1 June 2006, Science Council of Japan). The procedures were approved by the Institutional Review Board of Kanazawa University or the Laboratory Animal Ethics Committee of PhoenixBio.

## BrdU labelling in chimeric mice with humanized liver

Chimeric mice with humanized livers (PXB-mice, PhoenixBio) were generated from urokinase-type plasminogen activator-cDNA/ severe combined immunodeficiency mice transplanted with human hepatocytes, with approximately 80% of hepatocytes being humanized[48]. PXB-mice (male, >12 weeks, 15−21 g) were given a single subcutaneous injection at 5 mg kg$^{-1}$ for Fc(aMD4)B3 in 200 µl PBS or control PBS. Forty-four hours after administration of Fc(aMD4)B3 or control PBS, 5-bromo-2'-deoxyuridine (BrdU) (Sigma Chemical) was intraperitoneally injected into chimeric mice at a dose of 100 mg kg$^{-1}$ for 2 h before killing. Livers isolated from PXB-mice were treated with RNAlater for RNA-seq or prepared for BrdU staining. Formalin-fixed paraffin sections of chimeric livers were incubated with a rat anti-BrdU antibody at 3 µg ml$^{-1}$ (Abcam, ab6326), followed by an Alexa Fluor 488-conjugated anti-rat IgG goat polyclonal antibody at 1 µg ml$^{-1}$ (Abcam, ab150157). Nuclei were counterstained with 4′,6-diamidino-2-phenylindole (DAPI, Thermo Fisher) at 300 nM. Fluorescent images were obtained using Keyence BZ-X810. The number of BrdU-positive nuclei was determined by counting at least 15,000 cells in 20 randomly selected visual fields in sections from the lateral left lobe and medial right lobe per animal. The procedures involving humanized liver tissues were approved by the Utilization of Human Tissue Ethical Committee of PhoenixBio. All animal experimental procedures were conducted in accordance with the guidelines provided by the Proper Conduct of Animal Experiments (1 June 2006, Science Council of Japan). The procedures were approved by the Laboratory Animal Ethics Committee of PhoenixBio.

## Affinity to mTfR

MaxiSorp plates (Nunc) were coated with recombinant mouse TfR (2 µg ml$^{-1}$ in PBS, 50 µl per well, R&D Systems) at 4 °C overnight, washed with 0.05% Tween 20 in PBS and blocked with 1% BSA in PBS. Plates were incubated with 1:3 serial titrations of dTfR(aMD4), sTfR(aMD4) or Fc(aMD4) diluted with 0.05% Tween 20 and 1% BSA in PBS at room temperature for 1 h, washed with 0.05% Tween 20 in PBS, incubated with HRP-conjugated anti-human IgG Fc antibody (0.5 µg ml$^{-1}$, 100 µl per well, Bethyl Laboratories) at room temperature for 1 h and washed with 0.05% Tween 20 in PBS. Signals were generated by incubation with 3,3′,5,5′-tetramethylbenzidine substrate (Cell Signaling Technology) and absorbance was read using an ARVO MX plate reader (Perkin Elmer).

## BBB penetration

Wild-type C57BL/6 mice (female, 7–8 weeks, 17–20 g, obtained from SLC) were given a single tail-vein injection at equimolar amount per kg: 26.5 mg kg$^{-1}$ for dTfR(aMD4), 20.0 mg kg$^{-1}$ for sTfR(aMD4) or 12.0 mg kg$^{-1}$ for Fc(aMD4), diluted in 200 μl PBS. Twenty-four hours after injection, mice were anaesthetized and blood was drawn from the right ventricle and processed to serum. Brains were isolated from mice after perfusion with PBS at a rate of 2 ml min$^{-1}$ for 8 min. Each half of the brain was homogenized in 1% NP-40 (Calbiochem) in PBS containing CompleteMini EDTA-free protease inhibitor cocktail tablets (Roche Diagnostics), rotated at 4 °C for 1 h, spun at 14,000 r.p.m. for 20 min, and supernatants collected. The serum and the brain lysates were subjected to antibody measurement using a human immunoglobulin recognition immunoassay (human IgG ELISA quantitation set, Bethyl Laboratories). To calculate the concentrations in the brain, ng g$^{-1}$ brain was considered to be ng ml$^{-1}$. Each half of the brain was embedded in optimal cutting temperature compound (Sakura Finetek Japan), frozen in liquid nitrogen and sectioned at 10 μm thickness. After fixation in 4% paraformaldehyde in PBS for 30 min, the sections were blocked in 5% BSA, 0.3% Triton X-100 in PBS and stained with a goat anti-human IgG Fc antibody (1:500 dilution, Bethyl Laboratories) and a mouse anti-mouse NeuN antibody (1:500 dilution, Millipore), a mouse anti-GFAP antibody (1:500 dilution, GA5, Cell Signaling Technologies) or a mouse anti-Iba1 antibody (1:500 dilution, GT10312, GeneTex) in 1% BSA and 0.1% Triton X-100 in PBS at 4 °C overnight. Anti-human IgG Fc and anti-NeuN were visualized with an Alexa Fluor 488-conjugated anti-goat secondary antibody or an Alexa Fluor 594-conjugated anti-mouse secondary antibody, respectively (1:200 dilution, Thermo Fisher). Fluorescent images of cortical regions were obtained with the Keyence BZ-X810. This study was approved by the Institutional Review Board of Kanazawa University and performed in accordance with the guidelines and regulations.

## Statistical analysis

Graphpad Prism 6.0d was used for graphing and statistical analysis. Relevant statistical methods for individual experiments are detailed in figure legends.

## Reporting summary

Further information on research design is available in the Nature Research Reporting Summary linked to this article.

## Data availability

The main data supporting the findings of this study are available within the Article and its Supplementary Information. The RNA-seq data are available at the DDBJ Sequence Read Archive under accession numbers DRA014557 (hepatocyte spheroids) and DRA014558 (livers of PXB mice). The raw data generated during the study are available from the corresponding authors on reasonable request. Source data for the figures are provided with this paper.

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

## Acknowledgements

This work was supported by: World Premier International Research Center Initiative (WPI), MEXT, Japan; a Grant-in-Aid for JSPS Scientific Research (C) (20K06553) to K.S.; a Grant-in-Aid for JSPS Scientific Research (B) (19H03499) and Challenging Research (21K18250) to K.M.; Program for Basic and Clinical Research on Hepatitis from the Japan Agency for Medical Research and Development (AMED) (JP21fk0210087) to K.S.; AMED Platform Project for Supporting Drug Discovery and Life Science Research (Basis for Supporting Innovative Drug Discovery and Life Science Research) to H.S. (JP19am0101090) and J.T. (JP19am0101075); and a Grant-in-Aid for JSPS Scientific Research (B) (21H01771) to M.S. This work was performed under the Cooperative Research Program of the Institute for Protein Research, Osaka University (CR15-05), and an Extramural Collaborative Research Grant from the Cancer Research Institute (Kanazawa University). We thank Dolphin Co. Ltd for the English language review.

## Author contributions

K.S., H. Suga, J.T. and K.M. conceived and designed the study. K.S. performed expression and purification of proteins, cellular assays, SPR for FcRn, crosslink, RNA-seq, serum half-life experiments and mTfR-binding assay. N.S-N. performed flow cytometry and SPR for Met. E.M. performed a preliminary assessment of Fc grafts and their activities. S.W. performed protein expression and purification, and the mapping of the binding site via the chimeric Met experiments. N.M.R.-C. performed BBB penetration experiments. I.S. performed ds-variants. H. Sato performed immunohistochemistry and contributed to animal experiments. R.I. contributed to animal experiments. D.C.-C.V. contributed to animal experiments and critically revised the manuscript. C.Y. and C.T. performed BrdU labelling in PXB mice. M.S. performed HS-AFM. The manuscript was written by K.S. All authors discussed the results and commented on the manuscript.

## Competing interests

H. Suga and J.T. are co-founders and shareholders of MiraBiologics Inc. E.M., K.S. and K.M. are also shareholders of the same company. The other authors declare no competing interests.

## Additional information

**Extended data** is available for this paper at https://doi.org/10.1038/s41551-022-00955-6.

**Correspondence and requests for materials** should be addressed to Katsuya Sakai, Junichi Takagi or Kunio Matsumoto.

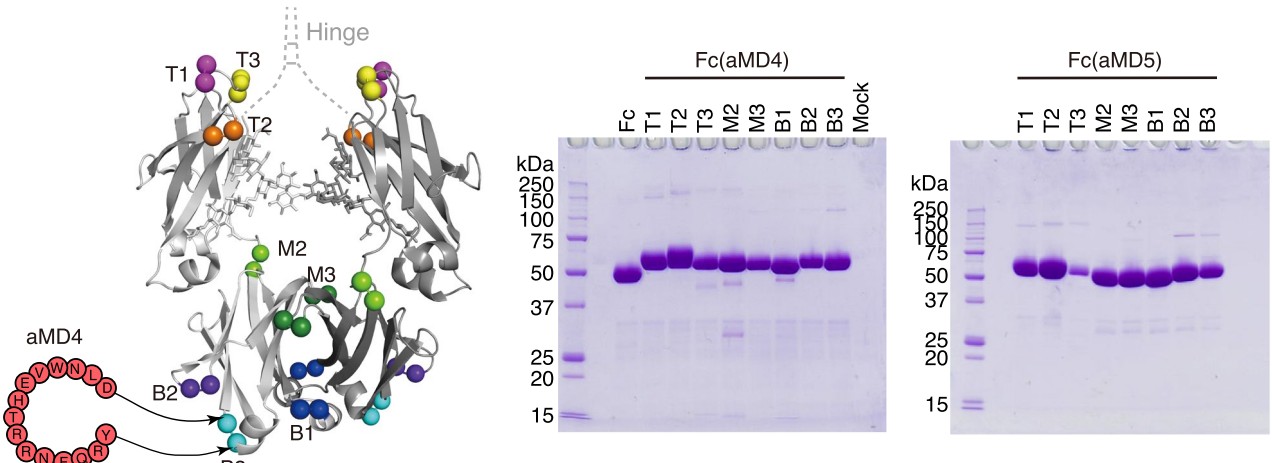

**Extended Data Fig. 1 | Expression of peptide-grafted Fc variants.** Conditioned medium of Expi293F suspension culture expressing control Fc or Fc variants were pulled down using protein A beads and analyzed by SDS-PAGE under non-reducing conditions and stained with Coomassie brilliant blue. The image for Fc(aMD4) was reproduced from our previous report[27].

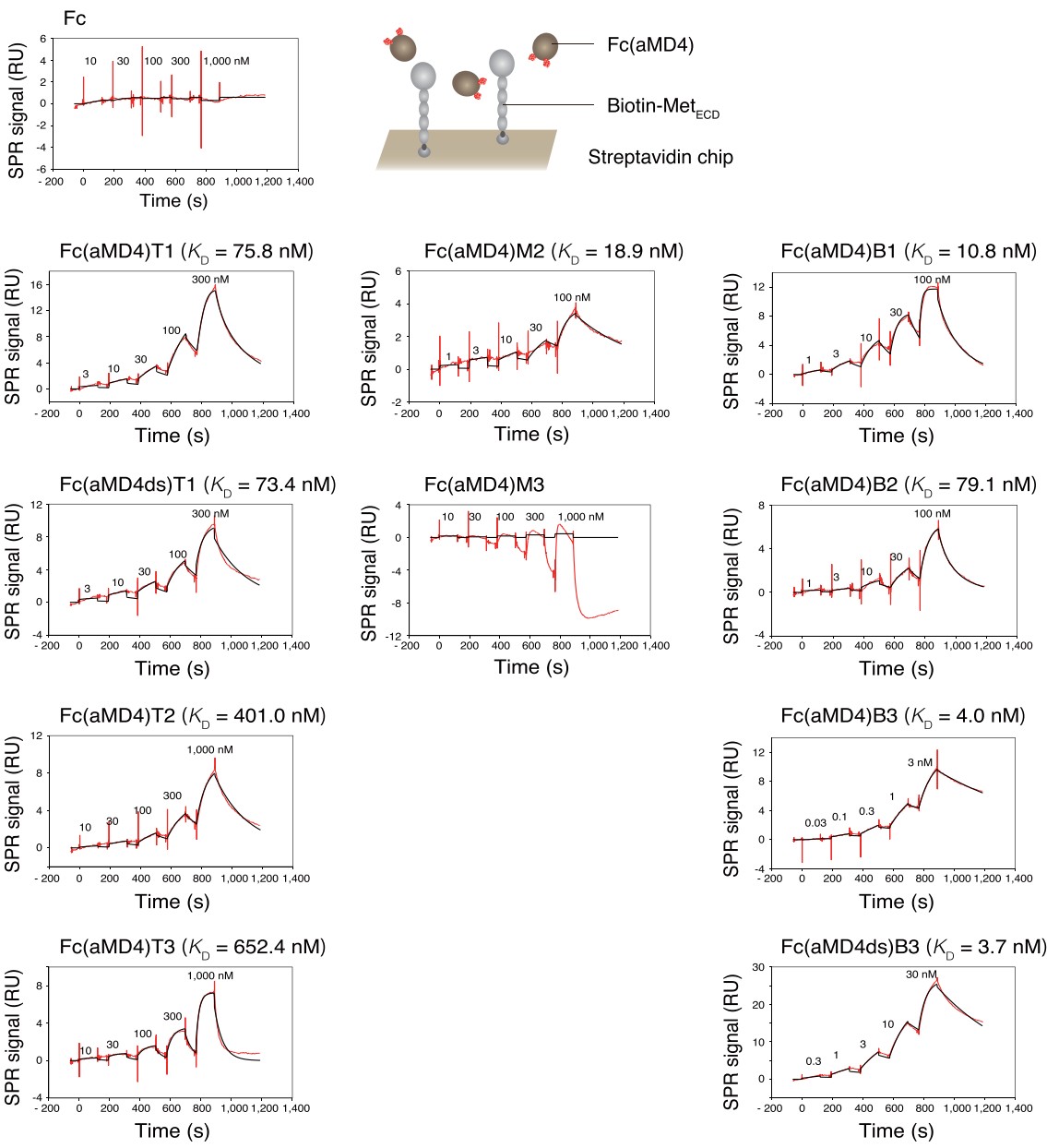

**Extended Data Fig. 2 | Binding of Fc or Fc(aMD4) to human Met$_{ECD}$.** Binding sensorgrams of Fc or Fc(aMD4) to human Met$_{ECD}$ as determined by surface plasmon resonance. Measured curves are shown in red. Fitted curves are shown in black. RU, resonance units; SPR, surface plasmon resonance.

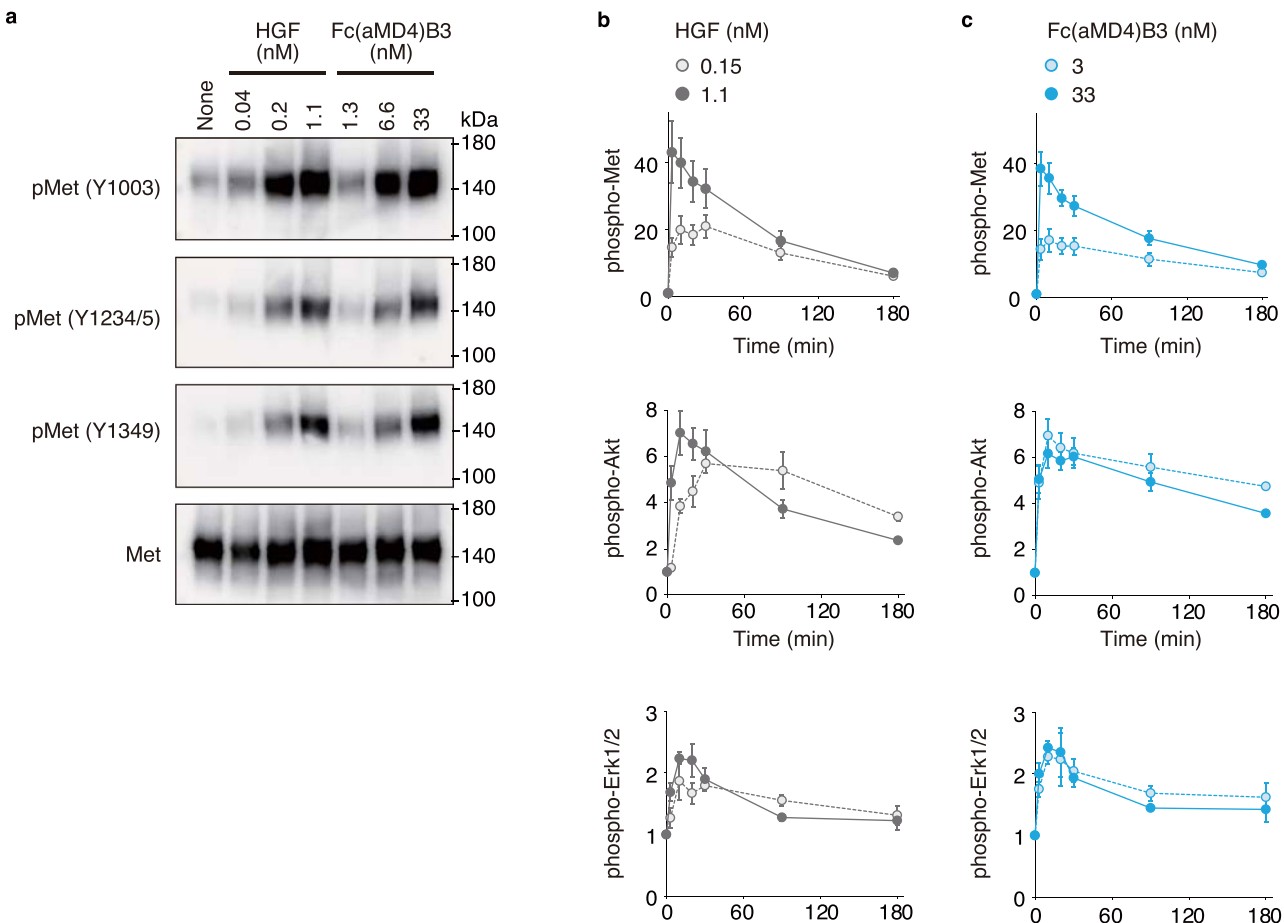

**Extended Data Fig. 3 | Met tyrosine phosphorylations (a) and induction kinetics (b, c) were comparable between Fc(aMD4)B3 and HGF. a**, Phosphorylation levels of Met tyrosine residues in EHMES-1 cells treated with HGF or Fc(aMD4)B3 for 10 min. The lysates were analyzed by Western blotting. **b-c**, EHMES-1 cells were treated with HGF (**b**) or Fc(aMD4)B3 (**c**). Activation of Met, Akt, or Erk1/2 at each time point after stimulation was quantified *in situ* with anti-phosphorylated Met (Tyr1234/1235), anti-phosphorylated Akt (S473), or anti-phosphorylated Erk1/2 (T202/Y204) antibody, respectively. The results are shown as the mean ± S.E.M. of fold-induction relative to non-stimulated controls (*n* = 4, independent experiments).

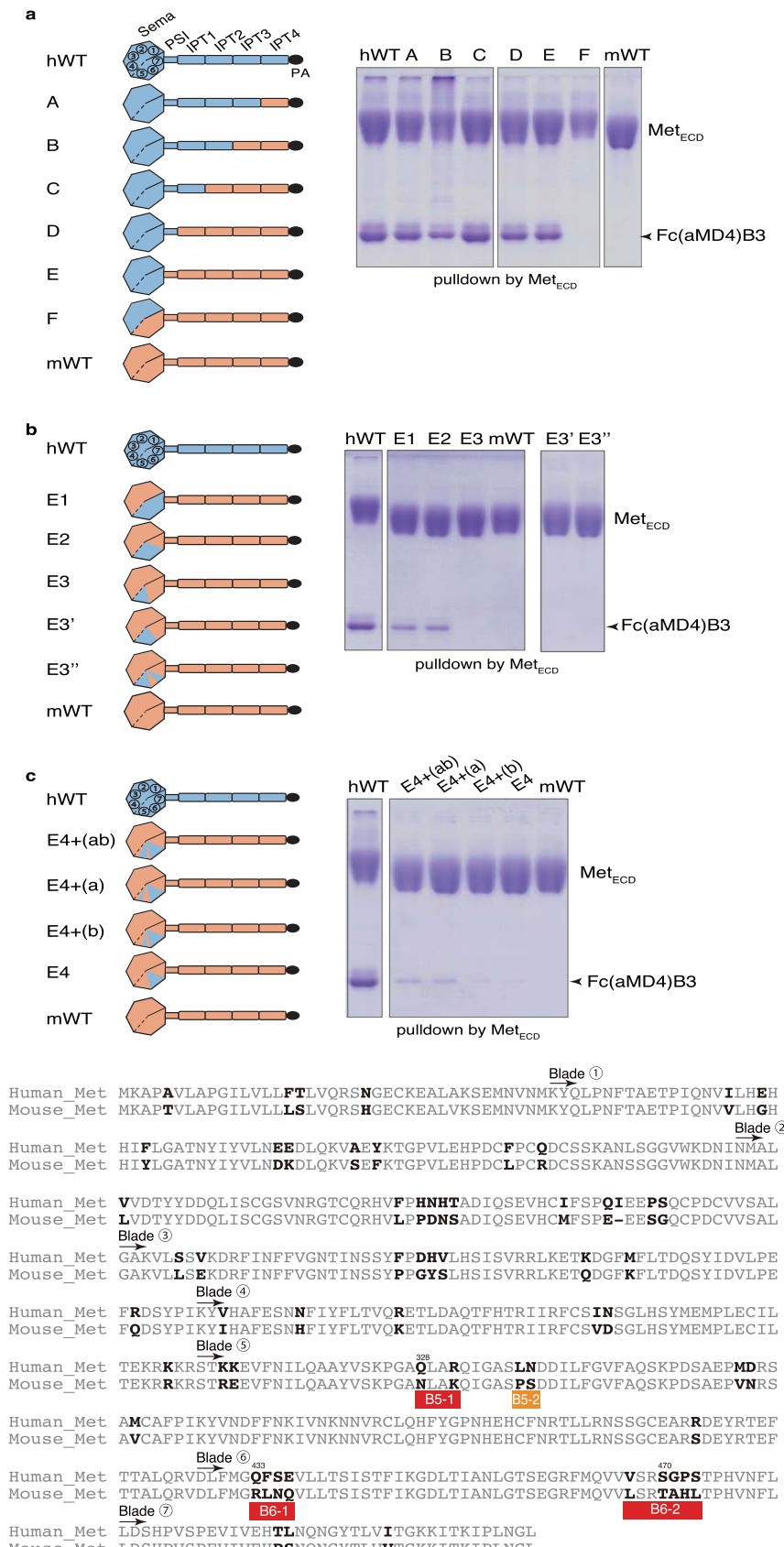

**d**

Human_Met  MKAP**A**VLAPGILVLL**FT**LVQRS**N**GECKEALAKSEMNVNMKY**G**QLPNFTAETPIQNV**I**LH**E**H
Mouse_Met  MKAP**T**VLAPGILVLL**LS**LVQRS**H**GECKEALVKSEMNVNMKYQLPNFTAETPIQNV**V**LH**G**H

Human_Met  HI**F**LGATNYIYVLN**EE**DLQKV**AEY**KTGPVLEHPDC**F**PC**Q**DCSSKANLSGGVWKDNINMAL
Mouse_Met  HI**Y**LGATNYIYVLN**DK**DLQKV**SEF**KTGPVLEHPDC**L**PC**R**DCSSKANSSGGVWKDNINMAL

Human_Met  **V**VDTYYDDQLISCGSVNRGTCQRHV**FP HNHT**ADIQSEVHC**I**FSP**QI**EE**PS**QCPDCVVSAL
Mouse_Met  **L**VDTYYDDQLISCGSVNRGTCQRHV**LP PDNS**ADIQSEVHC**M**FSP**E-**EE**SG**QCPDCVVSAL

Human_Met  GAKVL**SS V**KDRFINFFVGNTINSSY**FP DHV**LHSISVRRLKET**K**DGF**M**FLTDQSYIDVLPE
Mouse_Met  GAKVL**LSE**KDRFINFFVGNTINSSY**PP GYS**LHSISVRRLKET**Q**DGF**K**FLTDQSYIDVLPE

Human_Met  F**R**DSYPIKY**V**HAFESN**N**FIYFLTVQ**R**ETLDAQTFHTRIIRFCS**IN**SGLHSYMEMPLECIL
Mouse_Met  F**Q**DSYPIKY**I**HAFESN**H**FIYFLTVQ**K**ETLDAQTFHTRIIRFCS**VD**SGLHSYMEMPLECIL

Human_Met  TEKR**K**KRST**KK**EVFNILQAAYVSKPGA**Q**LA**R**QIGAS**LN**DDILFGVFAQSKPDSAEP**MD**RS
Mouse_Met  TEKR**R**KRST**RE**EVFNILQAAYVSKPGA**N**LA**K**QIGAS**PS**DDILFGVFAQSKPDSAEP**VN**RS

Human_Met  A**M**CAFPIKYVNDFFNKIVNKNNVRCLQHFYGPNHEHCFNRTLLRNSSGCEAR**R**DEYRTEF
Mouse_Met  A**V**CAFPIKYVNDFFNKIVNKNNVRCLQHFYGPNHEHCFNRTLLRNSSGCEAR**S**DEYRTEF

Human_Met  TTALQRVDLFMG**QFSE**VLLTSISTFIKGDLTIANLGTSEGRFMQVV**VS**R**SGPS**TPHVNFL
Mouse_Met  TTALQRVDLFMG**RLNQ**VLLTSISTFIKGDLTIANLGTSEGRFMQVV**LS**R**TAHL**TPHVNFL

Human_Met  LD**S**HPVSPEVIVEH**TL**NQNGYTLV**I**TGKKITKIPLNGL
Mouse_Met  LD**S**HPVSPEVIVEH**PS**NQNGYTLV**V**TGKKITKIPLNGL

**Extended Data Fig. 4 | See next page for caption.**

**Extended Data Fig. 4 | Mapping of Fc(aMD4)B3-binding sites on human Met$_{ECD}$. a**, The association between Fc(aMD4)B3 and human Met$_{ECD}$, mouse Met$_{ECD}$, or chimeric Met$_{ECD}$ that variably fused human and mouse was evaluated by pull-down assay. PA-tagged Met$_{ECD}$ was precipitated using anti-PA tag antibody conjugated beads. Consistent with our previous report that aMD4 peptide binds to human but not mouse Met$_{ECD}$[40], Fc(aMD4)B3 bound to human Met$_{ECD}$ (hWT) but not to mouse Met$_{ECD}$ (mWT). Replacement of the PSI domain and the IPT domains to mouse sequence (variants A–E) preserved the binding to Fc(aMD4)B3, indicating the PSI domain and the IPT domains were dispensable for Fc(aMD4)B3 binding. In contrast, human-specific residues in blades 5–7 of the Sema domain were required for Fc(aMD4)B3 binding (variant F). The numbers in the Sema domain indicate each blade. **b**, Human-specific residues in blade 6 (B6-1 and B6-2 in **d**) were indispensable for Fc(aMD4)B3 binding. **c**, Human-specific residues in blade 5 (B5-1 and B5-2 in **d**) were also indispensable or contributing for Fc(aMD4)B3 binding, respectively. **d**, Amino acid sequence alignment of human and mouse Met Sema domains. Residues that are different between human and mouse are indicated in **bold**. The red bars (B5-1, B6-1, and B6-2) indicate the residues indispensable for Fc(aMD4)B3 binding. The orange bar (B5-2) indicates the residues that contribute to Fc(aMD4)B3 binding.

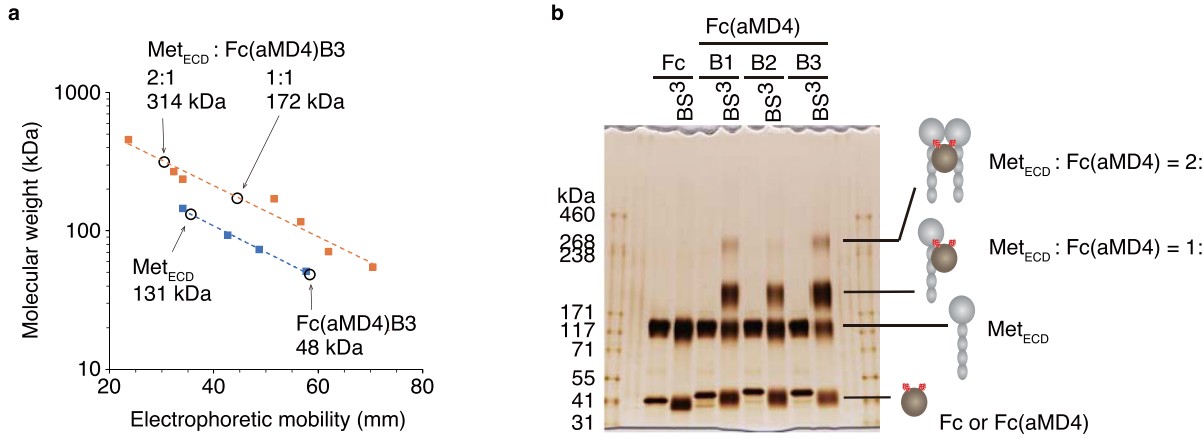

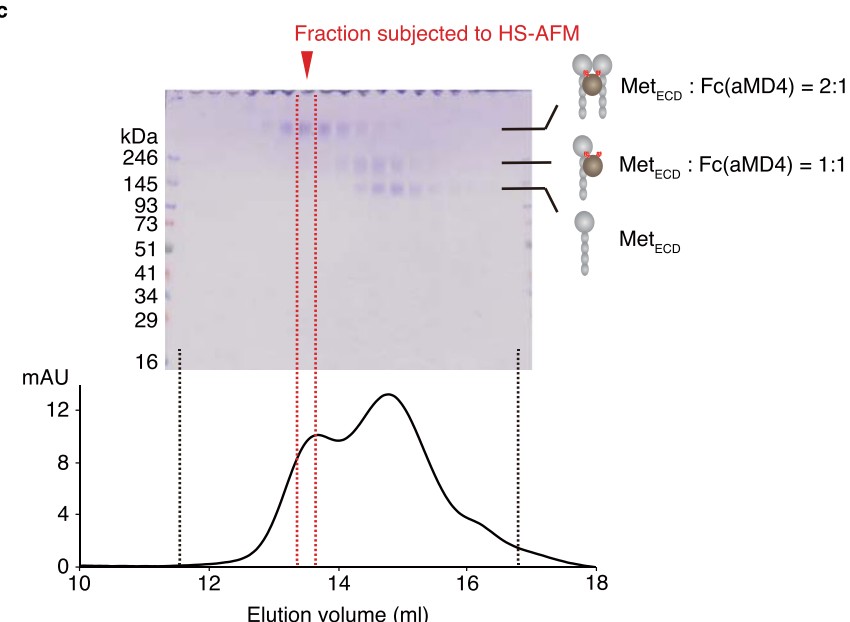

**Extended Data Fig. 5 | Analysis of BS³-stabilized 2:1 complex of Met$_{ECD}$ and Fc(aMD4)B3. a**, Molecular weight analyzed by SDS-PAGE indicated 2:1 and 1:1 complex of Met$_{ECD}$ and Fc(aMD4)B3. SDS-PAGE analysis was performed using 7.5% gels (blue, from Fig. 3b) and 3-10% gels (orange, from **b**). **b**, SDS-PAGE analysis of Met dimer formation by Fc(aMD4) in vitro. The complexes between Met$_{ECD}$ and Fc(aMD4) or control Fc were crosslinked by BS³ and analyzed by SDS-PAGE (3-10%

gel) under non-reducing conditions and silver stained. **c**, Met$_{ECD}$ and Fc(aMD4) B3 were crosslinked by BS³ and subjected to size exclusion chromatography (SEC). The results of non-reducing SDS-PAGE of the SEC fractions with Coomassie brilliant blue staining are shown. The fraction indicated in red was subjected to HS-AFM as shown in Fig. 3e, f.

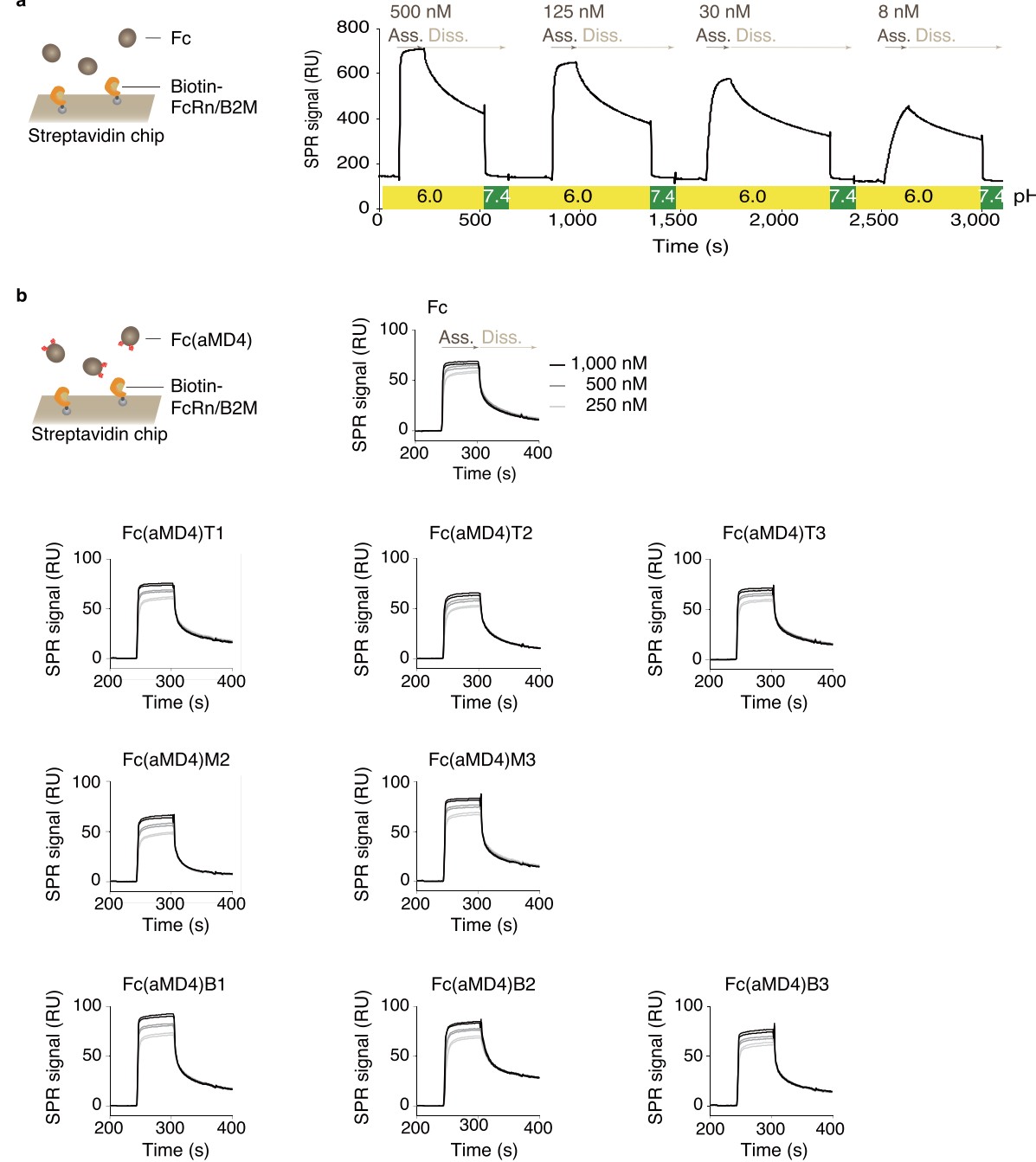

**Extended Data Fig. 6 | Binding of Fc or Fc(aMD4) to FcRn/β2 M for Fig. 4b. a**, Binding sensorgrams of Fc to FcRn/β2 M at pH 6.0 and 7.4. Ass, addition of Fc; Diss, buffer wash. **b**, Binding sensorgrams of Fc or Fc(aMD4) to FcRn/β2 M at pH 6.0. Sensorgrams show duplicates of three concentrations of analytes (250, 500, 1,000 nM). RU, resonance units; SPR, surface plasmon resonance.

|  | Corresponding author(s): | Katsuya Sakai, Junichi Takagi, and Kunio Matsumoto |
| --- | --- | --- |
|  | Last updated by author(s): | Sep 19, 2022 |

# Reporting Summary

## Statistics

For all statistical analyses, confirm that the following items are present in the figure legend, table legend, main text, or Methods section.

| n/a | Confirmed | |
| --- | --- | --- |
| ☐ | ☒ | The exact sample size (*n*) for each experimental group/condition, given as a discrete number and unit of measurement |
| ☐ | ☒ | A statement on whether measurements were taken from distinct samples or whether the same sample was measured repeatedly |
| ☐ | ☒ | The statistical test(s) used AND whether they are one- or two-sided *Only common tests should be described solely by name; describe more complex techniques in the Methods section.* |
| ☒ | ☐ | A description of all covariates tested |
| ☒ | ☐ | A description of any assumptions or corrections, such as tests of normality and adjustment for multiple comparisons |
| ☐ | ☒ | A full description of the statistical parameters including central tendency (e.g. means) or other basic estimates (e.g. regression coefficient) AND variation (e.g. standard deviation) or associated estimates of uncertainty (e.g. confidence intervals) |
| ☐ | ☒ | For null hypothesis testing, the test statistic (e.g. *F*, *t*, *r*) with confidence intervals, effect sizes, degrees of freedom and *P* value noted *Give P values as exact values whenever suitable.* |
| ☒ | ☐ | For Bayesian analysis, information on the choice of priors and Markov chain Monte Carlo settings |
| ☒ | ☐ | For hierarchical and complex designs, identification of the appropriate level for tests and full reporting of outcomes |
| ☒ | ☐ | Estimates of effect sizes (e.g. Cohen's *d*, Pearson's *r*), indicating how they were calculated |

*Our web collection on statistics for biologists contains articles on many of the points above.*

## Software and code

Policy information about availability of computer code

| Data collection | Chemiluminescence and absorbance were measured using an ARVO MX plate reader (Perkin Elmer). Cell-surface fluorescence was measured using an EC800 system (Sony). SPR was performed using Biacore T200 instrument (Cytiva) or Biacore 3000 instrument (Cytiva). Western blotting images were captured using a FUSION-SOLO.6S.EDGE (VIRVER). RNA-seq was performed using a DNBSEQ-G400 sequencer (MGI TECH). Immunohistochemical staining images were captured using a BZ-X810 (Keyence). LC-MS/MS was performed using a Nexera X2 UHPLC system (Shimadzu corporation) connected to a SCIEX TripleTOF 6600 system (AB SCIEX), HS-AFM images were collected using Igor Pro Ver. 6.3.6.0. (WaveMetrics). |
| --- | --- |
| Data analysis | Prism 6.0d (GraphPad) was used graphing and statistical analysis. Binding kinetics were analysed using Biacore T200 evaluation software version 3.2 (GE Healthcare) or BIAevaluation software version 4.1 (GE Healthcare). HS-AFM images were analysed using Igor Pro Ver. 6.3.6.0. (WaveMetrics). Flow-cytometry data were analysed by FlowJo software version 10.6.1. The LC-MS raw data were processed using SCIEX BioPharmaView software (SCIEX). RNA-seq data were analysed using cutadapt (ver. 1.9.1), sickle (ver 1.33), hisat2 software (ver. 2.2.0),and featureCounts (ver. 2.0.0). DEG analysis was performed using DEGES in TCC (ver. 1.18.0) and DESeq (ver. 1.30.0). HS-AFM images were analysed using Igor Pro Ver. 6.3.6.0. (WaveMetrics). Structures of Fc with aMD4 or aMD5 were predicted by ColabFold, AlphaFold2 using MMseqs2. CBB staining was quantified using IMAGEJ.JS. |

For manuscripts utilizing custom algorithms or software that are central to the research but not yet described in published literature, software must be made available to editors and reviewers. We strongly encourage code deposition in a community repository (e.g. GitHub). See the Nature Portfolio guidelines for submitting code & software for further information.

## Data

Policy information about availability of data

All manuscripts must include a data availability statement. This statement should provide the following information, where applicable:

- Accession codes, unique identifiers, or web links for publicly available datasets
- A description of any restrictions on data availability
- For clinical datasets or third party data, please ensure that the statement adheres to our policy

> The main data supporting the findings of this study are available within the Article and its Supplementary Information. The RNA-seq data are available at the DDBJ Sequence Read Archive under accession numbers DRA014557 (hepatocyte spheroids) and DRA014558 (livers of PXB-mice). Source data for the figures are provided with this paper. The raw data generated during the study are available from the corresponding authors on reasonable request.

# Field-specific reporting

Please select the one below that is the best fit for your research. If you are not sure, read the appropriate sections before making your selection.

☒ Life sciences ☐ Behavioural & social sciences ☐ Ecological, evolutionary & environmental sciences

For a reference copy of the document with all sections, see nature.com/documents/nr-reporting-summary-flat.pdf

# Life sciences study design

All studies must disclose on these points even when the disclosure is negative.

| | |
|---|---|
| Sample size | No statistical method was used to predetermine sample sizes. Sample sizes were chosen to establish statistical significance on the basis of similar experiments reported in the literature or of data from pilot experiments. Sample sizes were chosen as large as practically possible, and adequate statistics have been applied. |
| Data exclusions | No data were excluded from the analyses. |
| Replication | Reproducibility was tested through multiple inter-experimental and intra-experimental replicates, as described in the paper per each experiment. |
| Randomization | Mice were randomized into treatment groups. Each biochemical experiment in this study was rationally designed. Samples were not randomized for these experiments. |
| Blinding | Blinding was not used, because the design, execution and analysis of certain experiments was in many cases performed by a single investigator. This was necessary for data analysis and to minimize potential transposition error. For RNA-seq and LC-MS/MS analyses, sample preparation, and data acquisition and analysis were outsourced, without prior knowledge of the results. |

# Reporting for specific materials, systems and methods

We require information from authors about some types of materials, experimental systems and methods used in many studies. Here, indicate whether each material, system or method listed is relevant to your study. If you are not sure if a list item applies to your research, read the appropriate section before selecting a response.

## Materials & experimental systems

| n/a | Involved in the study |
|---|---|
| ☐ | ☒ Antibodies |
| ☐ | ☒ Eukaryotic cell lines |
| ☒ | ☐ Palaeontology and archaeology |
| ☐ | ☒ Animals and other organisms |
| ☒ | ☐ Human research participants |
| ☒ | ☐ Clinical data |
| ☒ | ☐ Dual use research of concern |

## Methods

| n/a | Involved in the study |
|---|---|
| ☒ | ☐ ChIP-seq |
| ☐ | ☒ Flow cytometry |
| ☒ | ☐ MRI-based neuroimaging |

## Antibodies

| | |
|---|---|
| Antibodies used | Anti-phosphorylated Erk1/2 (T202/Y204) (1:1,000 dilution, D13.14.4E, Cell Signaling Technology). Anti-Erk1/2 (1:1,000 dilution, 137F5, Cell Signaling Technology). Anti-phosphorylated Akt (S473) antibody (1:1,000 dilution, D9E, Cell Signaling Technology). Anti-Akt antibody (1:1,000 dilution, 11E7, Cell Signaling Technology). Anti-GAPDH antibody (1:1,000 dilution, 14C10, Cell Signaling Technology). |

Anti-Met antibody (Immunoprecipitation, 1 µg for 200 µl lysates, D-4, Santacruz).
Anti-Met antibody (Western bloting, 1:1,000 dilution, D1C2, Cell Signaling Technology).
Abti-phosphorylated Met antibody (Y1234/Y1235) (1:1,000 dilution, D26, Cell Signaling Technology).
Abti-phosphorylated Met antibody (Y1003) (Western bloting, 1:1,000 dilution, 13D11, Cell Signaling Technology).
Abti-phosphorylated Met antibody (Y1349) (Western bloting, 1:1,000 dilution, 130H2, Cell Signaling Technology).
HRP-conjugated anti-rabbit Immunoglobulin antibody  (1:2000 dilution for western blotting, 1:1000 dilution for cellular Met phosphorylation, #P0448, Dako).
HRP-conjugated anti-mouse Immunoglobulin antibody  (1:2000 dilution for western blotting, #P0447, Dako).
Alexa Fluor 488-labeled anti-human-IgG used for flow cytometry (1:400 dilution, #A11013, ThermoFisher Scientific).
HRP-conjugated anti-human IgG Fc antibody used for ELISA (0.5 µg/ml, 100 µl/well, Bethyl Laboratories).
Anti-human IgG Fc antibody used for IHC (1: 500 dilution, Bethyl Laboratories).
Anti-mouse NeuN antibody used for IHC (1: 500 dilution, Millipore).
Alexa Fluor 488-conjugated anti-goat secondary antibody used for IHC (1:200 dilution, Thermo Fisher Scientific).
Alexa Fluor 594-conjugated anti-mouse secondary antibody used for IHC (1:200 dilution, Thermo Fisher Scientific).
Anti-human HGF rabbit polyclonal antibody for ELISA (10 µg/mL for coating, 2 µg/mL for detection, In house prepared).
Anti-BrdU antibody used for IHC (3 µg/ml, ab6326, Abcam).
Alexa Fluor 488-conjugated anti-rat secondary antibody used for IHC (1 µg/ml, ab150157, Abcam).
Anti-PA tag antibody (NZ-1) (150 µl NZ1 Sepharose for pulldown, FUJIFILM Wako).
Anti-GFAP antibody used for IHC (1:500 dilution, GA5, Cell Signaling Technology).
Anti-Iba1 antibody used for IHC (1:500 dilution, GT10312, GeneTex).

Validation

All the antibodies were optimized and validated (per assay and species) by the supplier. Information on any validation statements can be found on the manufacturer's website.

The anti-human HGF rabbit polyclonal antibody for ELISA was prepared as described in; Suzuki, Y. et al. Inhibition of Met/HGF receptor and angiogenesis by NK4 leads to suppression of tumor growth and migration in malignant pleural mesothelioma. Int J Cancer. 127, 1948-1957 (2010), and was characterized via ELISA, as described in Jangphattananont, N. et al. Distinct Localization of Mature HGF from its Precursor Form in Developing and Repairing the Stomach. Int. J. Mol. Sci. 20 (2019).

The anti-PA tag antibody NZ-1 was characterized in Fujii, Y. et al. PA tag: A versatile protein tagging system using a super high affinity antibody against a dodecapeptide derived from human podoplanin. Protein. Expres. Purif. 95, 240-247 (2014).

# Eukaryotic cell lines

Policy information about cell lines

Cell line source(s)

EHMES-1 was provided by Dr. Hamada (Ehime University, Ehime, Japan). EHMES-1 was established as described in Yokoyama, A. et al. Origin of heterogeneity of interleukin-6 (IL-6) levels in malignant pleural effusions. Oncol Rep. 1,507-511 (1994).
CHO-K1 was obtained from ATCC (RRID:CVCL_0214).
Expi293F was obtained from Thermo Fisher (RRID: CVCL_D615).
Met-knockout CHO-K1 cells and Met-reconstituted CHO-K1 cells were described in ref. 27.

Authentication

All cell lines were expanded upon initial receipt to create large stocks of frozen vials. To limit the risks of cross-contamination and over-subculturing, all cell lines used in our laboratory are passaged for no more than one month. Cell morphology is monitored several times per week during culture. If there are any changes in cell appearance, growth kinetics or performance in routine assays in our laboratory, cells are discarded and a new culture is established from frozen stocks.

Mycoplasma contamination

The cell lines used tested negative for mycoplasma.

Commonly misidentified lines
(See ICLAC register)

No commonly misidentified cell lines were used.

# Animals and other organisms

Policy information about studies involving animals; ARRIVE guidelines recommended for reporting animal research

Laboratory animals

C57BL/6 mice were purchased from Japan SLC.

mFcRn-/- mice (stock number: 003982) and hFcRn Tg 276 homozygote mice (stock number: 004919) were purchased from The Jackson Laboratory, and bred in order to obtain mFcRn-/-, hFcRn Tg 276 heterozygote mice.

Chimaeric mice with humanized livers (PXB-mice, PhoenixBio) were generated from urokinase-type plasminogen activator-cDNA/ severe combined immunodeficiency mice transplanted with human hepatocytes with approximately 80% of hepatocytes being humanized (ref. 48).

Mice were bred and maintained at the Advanced Science Research Center Institute for Experimental Animals, Kanazawa University, or at PhoenixBio Inc.

Wild animals

The study did not involve wild animals.

Field-collected samples

The study did not involve samples collected from the field.

Ethics oversight

All animal experimental procedures were conducted in accordance with the guidelines provided by the Proper Conduct of Animal Experiments (June 1, 2006; Science Council of Japan). The procedures were approved by the Institutional Review Board of Kanazawa

University or Laboratory Animal Ethics Committee of PhoenixBio. The procedures involving humanized liver tissues were approved by Utilization of Human Tissue Ethical Committee of PhoenixBio.

Note that full information on the approval of the study protocol must also be provided in the manuscript.

# Flow Cytometry

## Plots

Confirm that:

☒ The axis labels state the marker and fluorochrome used (e.g. CD4-FITC).

☒ The axis scales are clearly visible. Include numbers along axes only for bottom left plot of group (a 'group' is an analysis of identical markers).

☒ All plots are contour plots with outliers or pseudocolor plots.

☒ A numerical value for number of cells or percentage (with statistics) is provided.

## Methodology

| | |
|---|---|
| Sample preparation | The binding of peptide-grafted Fc proteins to Met-knockout CHO cells and Met-reconstituted CHO cells (ref. 27) were detected using flow cytometry. Cells were detached from dishes by a brief treatment with 0.025% trypsin and 1 mM ethylenediaminetetraacetic acid, plated at 200,000 cells per well and incubated with peptide-grafted Fc proteins diluted at 10 µg/ml in 100 µl Ham's F-12 medium containing 5% FBS on ice for 1.5 h. After washing twice with ice-cold PBS, cells were incubated with Alexa Fluor 488-labeled goat anti-human IgG (1:400 dilution in 100 µl Ham's F-12 medium containing 5% FBS, Thermo Fisher Scientific, A11013) on ice for 30 min, then analysed on an EC800 system (Sony). The data were analysed with FlowJo software (Tomy Digital Biology). |
| Instrument | EC800 system (Sony). |
| Software | FlowJo software (Tomy Digital Biology). |
| Cell population abundance | the purity of gated populations was generally in the range of 60–90%, as shown in Supplementary Fig. 11. |
| Gating strategy | Gate on fsc vs. ssc was set to include all cell populations, but excluding debris and dead cells as shown in Supplementary Fig. 11. |

☒ Tick this box to confirm that a figure exemplifying the gating strategy is provided in the Supplementary Information.

