## [Peer Review File · Nature Biomedical Engineering]

Designing receptor agonists with enhanced pharmacokinetics by grafting macrocyclic peptides into fragment crystallizable regions

Corresponding author: Katsuya Sakai

Editorial note

This document includes relevant written communications between the manuscript's corresponding author and the editor and reviewers of the manuscript during peer review. It includes decision letters relaying any editorial points and peer-review reports, and the authors' replies to these (under 'Rebuttal' headings). The editorial decisions are signed by the manuscript's handling editor, yet the editorial team and ultimately the journal's Chief Editor share responsibility for all decisions.

Any relevant documents attached to the decision letters are referred to as **Appendix #**, and can be found appended to this document. Any information deemed confidential has been redacted or removed. Earlier versions of the manuscript are not published, yet the originally submitted version may be available as a preprint. Because of editorial edits and changes during peer review, the published title of the paper and the title mentioned in below correspondence may differ.

The rebuttal letters included a few figures that were marked as 'for review only'. These figures have been removed from this document.

Correspondence

Sat 05 Feb 2022

Decision on Article nBME-21-2187

Dear Dr Sakai,

Thank you again for submitting to *Nature Biomedical Engineering* your manuscript, "Lasso-grafting Fc with macrocyclic peptides to create designer receptor agonists", and apologies for the substantial delay in providing the feedback. The manuscript has been seen by three experts, yet one expert has failed to provide their report, despite our regular chasing messages. You will find the reports of two reviewers at the end of this message. You will see that the reviewers appreciate the work, and that they raise a number of technical criticisms that we hope you will be able to address. In particular, we would expect that a revised version of the manuscript provides:

- * Enhanced description, with supporting evidence, of the lasso-grafting mechanism, as per the relevant queries of the reviewers.
- * Thorough methodological details.

When you are ready to resubmit your manuscript, please upload the revised files, a point-by-point rebuttal to the comments from all reviewers, the reporting summary, and a cover letter that explains the main improvements included in the revision and responds to any points highlighted in this decision.

Please follow the following recommendations:

- * Clearly highlight any amendments to the text and figures to help the reviewers and editors find andunderstand the changes (yet keep in mind that excessive marking can hinder readability).

* If you and your co-authors disagree with a criticism, provide the arguments to the reviewer (optionally, indicate the relevant points in the cover letter).

* If a criticism or suggestion is not addressed, please indicate so in the rebuttal to the reviewer comments and explain the reason(s).

* Consider including responses to any criticisms raised by more than one reviewer at the beginning of the rebuttal, in a section addressed to all reviewers.

* The rebuttal should include the reviewer comments in point-by-point format (please note that we provide all reviewers with the reports as they appear at the end of this message).

* Provide the rebuttal to the reviewer comments and the cover letter as separate files.

We hope that you will be able to resubmit the manuscript within 15 weeks from the receipt of this message. If this is the case, you will be protected against potential scooping. Otherwise, we will be happy to consider a revised manuscript as long as the significance of the work is not compromised by work published elsewhere or accepted for publication at *Nature Biomedical Engineering*.

We hope that you will find the referee reports helpful when revising the work, which we look forward to receive. Please do not hesitate to contact me should you have any questions.

Best wishes,

Pep

Pep Pàmies
Chief Editor, Nature Biomedical Engineering

Reviewer #2 (Report for the authors (Required)):

Building on their previous works, the current study by Sakai and colleagues shows that the functional insertion of macrocyclic peptide pharmacophores into the protein loops of an immunoglobulin Fc or antibody leads to Fc- or antibody-based receptor agonists with long circulatory residency and BBB penetrance. Although this protein engineering methodology termed “lasso-grafting” was reported on in principal previously by the same team, the current findings go significantly beyond the de novo discovery of lasso-grafting (NCOMMS 2021), a macrocyclic peptide-based inhibition and imaging approach of HGF (Nat Chem Biol 2019), and a macrocyclic peptide HGF-agonistic grafting approach to a self-assembling protein nanocapsule (iScience 2021), as a specific approach is presented to generate a MET receptor agonist by grafting HTS-identified macrocyclic HGF-mimicking peptides into Fc- or transferrin receptor (TfR) antibody loops to come up with receptor agonists that have a long serum half-life and very promising BBB penetrance properties. They show that the grafted Fc constructs are able to dimerize the receptor MET, trigger receptor signaling responses comparable to the natural ligand HGF, while lasso-grafting did not compromise affinity of the grafted Fc construct for the neonatal Fc receptor (FcRn) that is responsible for the favorable PK properties and allowed for anti-TfR passaging into the brain parenchyma following systemic administration at a brain/circulation ratio of roughly 10%. Although not entirely new, the current paper is only one of very few reports of successful grafting of de novo identified peptides onto natural protein scaffolds, here Fc, without any apparent adverse effects on Fc loop folding. They also identified several suitable insertion points in Fc loop structures, demonstrating that the method is flexible and tunable.

Thus, the claim and conclusion that lasso-grafting of macrocyclic peptides into protein scaffolds with specific (preserved) properties is a paradigm for engineering designer receptor agonists (here MET agonists) seems justified.

Major technical criticisms or questions

1. Methods

The expression and purification of Met binder lasso-grafted Fc variants was verified by SDS-PAGE in combination with Coomassie staining. I would suggest to confirm "purity" by silver staining. In fact, what is the degree of purity estimated to be? 80%, 90%?

2. Ext Fig 3: The data on the disulphide-bonded derivatives is not fully convincing. How can authors be sure that the ds bridge is actually closed in the context of the Fc holo-molecule? The very slight shift in electrophoretic mobility could very well also come from the insertion of the cysteines as such. How about doing a confirmatory experiment with a corresponding isosteric double-serine variant?

Results

3. Western blot in Fig. 1d could be improved; phospho-AKT etc. bands run funny (smiley shape)

4. How can it be explained that for aMD5, B1 was the best grafting site, whereas it is B3 for aMD4? Are there any conspicuous residues in the sequences that could explain this difference? I would suggest to run sequence comparisons (hydrophobicity, charges, bulkiness of residues etc.) that could give an insight, and perhaps include such bioinformatic data as a Suppl. Table.

5. The phospho-Met WB of the primary hepatocytes in Fig. panel 1d is not very convincing; please consider running this again or switch to a "better" anti-phospho-Ab.

6. The transcriptomics analysis performed on the human hepatocyte spheroids is very valuable! Can authors comment on the (few) differences observed between HGF treatment and the Fc variants? For example, FGF19, SCL2A. Any explanation, why in some cases the duplicates differ more than the different treatment regimens?

7. Comment/appraisal: The Met-KO versus reconstituted CHO-Met approach is very elegant and offers strong evidence for binding.

8. I am a bit puzzled by the data in Figure 2A and Extended Data Fig. 4. The mapping basically suggests that the agonistic Fc variants bind to a different site on Met than HGF. This would imply that the lasso-grafts behave as allosteric agonists... with all implied pros and cons. Could the authors comment on this?

9. Figure 2B: from the molecular weight markers that were co-electrophoresed, it is hard to actually claim that the highest band corresponds to a 2:1 complex. 2:2 might also be possible. But of course, in conjunction with the super-cool AFM experiments, the conclusion appears justified. Still, authors should consider re-doing the electrophoresis, e.g. using another gel (gradient gel, low percentage etc.).

10. It would be interesting, in Figure 4, to counterstain the brain sections for an astroglia (GFAP) and a microglia (Iba1) marker, because of potential applications for glioblastoma and astrocytoma. What about the Met receptor? Is that expressed in brain?

Minor criticisms and comments

1. Discussion

Authors state "One major drawback of current BBB permeating antibodies is that their low brain-to-serum ratio and long half-life in circulation would often result in systemic concentrations that exceed the desired range for extended periods of time." Are they speculating about the BBB permeation of antibodies such as Aducanumab? If so, how could the current lasso-grafting approach come into play?

2. Abstract and general

The term "thioether"-macrocylic peptide should be avoided or down-toned, as in the end not the initially screened-for thioether peptide is used but a linearized or ds variant.

3. Introduction

When it comes to therapeutic targeting, the focus typically is on inflammatory or pro-cancerogenic cytokines such that the approach then is a "blocking" one. Exceptions next to HGF (which rather is classified as a growth factor) would be IL-10 and perhaps TGF β . Accordingly, there is a limited number of cytokine-based

“agonist” approaches. Maybe, authors could expand their discussion a bit more and list and discuss potential other “agonistic” targets to further underscore the potential utility of the here presented lasso-grafting approach.

4. Results

Although published previously, the actual “grafting” principle should be described better.

This reviewer has a biochemistry background, but I had to go back to virtually all previous publications in NCOMMS, Nat Chem Biol, and Angewandte to be able to fully understand how this works. To the end, how important is the thioether bond present in the initially identified Met-binders? Along the same lines, what changes may be induced by the spacer glycine-residues?

5. typo “yeilding” in methods

Reviewer #3 (Report for the authors (Required)):

In this manuscript, Sakai and colleagues describe thioether-macrocytic peptide pharmacophores grafted into Fc-receptor scaffold to generate therapeutic proteins with pharmacological activity in vivo. This work builds on a platform recently developed by this group for selecting and stabilizing macrocytic peptides by grafting into a protein scaffold (lasso-grafting) (Mihara et al. Nature Comms 2021, ref 21) In this work, lasso-grafted FcR with Met receptor ligands are optimized further for high-affinity binding to c-Met. In addition to showing activation of the Met receptor in cells stimulated ex-vivo, the authors study the in vivo properties of these lasso-grafted molecules in mice engrafted with human hepatocytes. As expected from the Fc scaffold, the grafts displayed extended half-lives of ~50 hours compared to the < 1 hour half-life of HGF, with the ability to activate hepatocyte proliferation after a single injection. Grafting of similar cyclic peptides onto an anti-transferrin receptor antibody conferred significant brain penetration, particularly with a single-arm TfR configuration

Overall this paper represents an advance in understanding and translatability of this potential therapeutic platform over the group’s previous work, with the first demonstration of in vivo PD with this type of graft and use of the known ability of anti-transferrin receptor antibody to endow brain penetrance. However, there are a number of specific issues that should be addressed before publication

Major

1. Figure 1: The authors claim identical signaling for the lasso grafts but only show a single time point. Other time points should be studied for pERK or pMet also at lower doses given the slower on rate if not done in their previous work. As the Lasso graft does not compete with HGF – could that activate different phosphorylation sites on cMet? Have they done any mass spec to probe this in an objective way?

2. Figure 2: Met dimerization induced by the aMD4 macrocycle graft that is shown should be compared to HGF induced dimerization. The relative ability of the lasso grafts vs HGF to induce receptor dimerization is not clear from data presented.

3. Figure 4: Although one of the more novel aspects of the paper, some of the brain penetration data are not clear and fall short of the rigor of the other experiments:

a. Why are different mg/kg doses used for the different anti-TfR fusions (line 312)?

b. In addition to the brain/serum concentration ratio, a brain/body ratio should be calculated to understand the biodistribution more globally. This value will help clarify what percentage of the molecule enters the brain relative to the dose administered in a manner that allows comparison with other technologies. The typical levels reported for other technologies are 1-2% entering the brain, which is in line with the weight ratio of the brain to body.

c. Please explain the rationale for selecting the 24 hour time point for the brain experiments.

d. The biological activity of the Met agonists in the brain could not be tested in vivo as the target is not in the CNS and the macrocycles are not cross-reactive to mouse. For example in their previous paper the authors develop a binder to PlxnB1 which is a CNS-expressed gene

4. Discussion: The authors should compare the merits of this novel format vs antibody-based biologics such as bispecific anti-TfR/anti-cytokine receptor agonists both for therapeutic value and in terms of potential

immunogenicity. Is there any data on immunogenicity of the lasso grafts?

Minor

1. In Figure 1, in addition to the heat map shown, a dot plot should be shown for gene induction in response to HGF vs Fc(aMD4) to better appreciate the correlation between the gene expression changes induced by the two ligands
2. The authors should cite <https://www.ncbi.nlm.nih.gov/pmc/articles/PMC7849817/> and discuss their transferrin affinity in that context (they used an older paper)

Sat 16 Jul 2022

Decision on Article NBME-21-2187A

Dear Dr Sakai,

Thank you for your patience in waiting for the feedback on your revised manuscript, "Lasso-grafting Fc with macrocyclic peptides to create designer receptor agonists". Having consulted with the original reviewers and one more expert (Reviewer #4; whose comments you will also find at the end of this message), I am pleased to write that we shall be happy to publish the manuscript in *Nature Biomedical Engineering*.

We will be performing detailed checks on your paper and will send you a checklist detailing our editorial and formatting requirements in due course.

In the meantime, please consider the comments of Reviewers #1 and #4 regarding the clarity of Extended Fig. 3; and, optionally, a brief discussion of the suitability of the agonist for potentially treating liver disease.

Best wishes,

Pep

Pep Pàmies
Chief Editor, Nature Biomedical Engineering

Reviewer #2 (Report for the authors (Required)):

Authors offer an extensively and carefully revised manuscript and, overall, have satisfactorily responded to my critique.

Regarding my second point (How can authors be sure that the ds bridge is actually closed in the context of the Fc holo-molecule?), I am fine with their approach of running reduced versus non-reduced gels plus LC-MS digests. The results convincingly show that the disulfide has formed for T1, B2, and B3. And I agree that with these data, the analysis of a double serine mutant wouldn't be necessary. The indicated molecular weights in new Extended Figure 3b are, however, a bit confusing for the reader. I assume the non-reducing panel refers to an Fc dimer, while the reducing one to a monomer. I would recommend to somehow indicate and clarify this in the figure. Alternatively, full size gels from 10 100 kD or so could be shown for both conditions.

Thank you for clarifying as to my point #8. I actually didn't think of true "allostery" with the dimerizers binding to Met "in addition" to the endogenous agonist, but actually had used the word "allosteric" just to indicate binding at a different site compared to HGF. I think the authors explanations have clarified my point, although, I have seen this right, Reviewer # 3 had a similar issue.

Reviewer #3 (Report for the authors (Required)):

I have reviewed the authors' point-by-point response and their response fully addresses the concerns. Thank you for carefully addressing them, and my apologies that this re-review took some time.

Reviewer #4 (Report for the authors (Required)):

This is an elegant study that marries the advantages of a Fc domain with peptides that bind to an epitope. The authors have addressed the comments that were raised by the reviewers. While this builds on the previous studies from the same group on this platform, the fact that here they show Met agonism with the platform is extremely relevant and could have high impact. This is because MET agonists are needed as therapeutics. Data presented is strong from a biochemical and signaling perspective. The use of humanized liver implants is also elegant.

While the authors show brain delivery, this is a distraction from what could be a potential translation of the product. HGF cannot be used as a therapeutic as it degrades in circulation (which could also explain the PK data in this paper) and its heparin binding limits its access into tissues. A potential therapeutic use of this agonist vs HGF could have been in NASH models or in models of liver failure. The prolonged circulation time of the agonist, and potentially absence of heparin binding (need to test for that) could have given a significant therapeutic advantage over HGF. Efficacy in a disease model would significantly increase the impact.

Rebuttal 1

Point-By-Point Response to the Reviewers' Comments

On 6th February 2022, we received comments from two reviewers, along with a decision by the editors. We deeply appreciate the time and effort each of the reviewers have dedicated to carefully evaluating our manuscript and providing insightful feedbacks aimed to strengthen the quality of our paper. We have taken the comments from all reviewers seriously and carefully planned additional experiments and analyses to address them. In the following paragraphs, we respond to the comments made by each reviewer. Our responses are in black while the reviewers' comments are in blue and italicized.

Reviewer #2 (Report for the authors (Required)):

Building on their previous works, the current study by Sakai and colleagues shows that the functional insertion of macrocyclic peptide pharmacophores into the protein loops of an immunoglobulin Fc or antibody leads to Fc- or antibody-based receptor agonists with long circulatory residency and BBB penetrance. Although this protein engineering methodology termed "lasso-grafting" was reported on in principal previously by the same team, the current findings go significantly beyond the de novo discovery of lasso-grafting (NCOMMS 2021), a macrocyclic peptide-based inhibition and imaging approach of HGF (Nat Chem Biol 2019), and a macrocyclic peptide HGF-agonistic grafting approach to a self-assembling protein nanocapsule (iScience 2021), as a specific approach is presented to generate a MET receptor agonist by grafting HTS-identified macrocyclic HGF-mimicking peptides into Fc- or transferrin receptor (TfR) antibody loops to come up with receptor agonists that have a long serum half-life and very promising BBB penetrance properties. They show that the grafted Fc constructs are able to dimerize the receptor MET, trigger receptor signaling responses comparable to the natural ligand HGF, while lasso-grafting did not compromise affinity of the grafted Fc construct for the neonatal Fc receptor (FcRn) that is responsible for the favorable PK properties and allowed for anti-TfR passaging into the brain parenchyma following systemic administration at a brain/circulation ratio of roughly 10%. Although not entirely new, the current paper is only one of very few reports of successful grafting of de novo identified peptides onto natural protein scaffolds, here Fc, without any apparent adverse effects on Fc loop folding. They also identified several suitable insertion points in Fc loop structures, demonstrating that the method is flexible and tunable.

Thus, the claim and conclusion that lasso-grafting of macrocyclic peptides into protein scaffolds with specific (preserved) properties is a paradigm for engineering designer receptor agonists (here

MET agonists) seems justified.

Response: We deeply appreciate the reviewer's effort to evaluate our work and provide insightful feedback that strengthens the quality of our paper. We followed the reviewer's suggestions and perform experiments to address them.

Major technical criticisms or questions

1. Methods

The expression and purification of Met binder lasso-grafted Fc variants was verified by SDS-PAGE in combination with Coomassie staining. I would suggest to confirm "purity" by silver staining. In fact, what is the degree of purity estimated to be? 80%, 90%?

Response: As requested by the reviewer, we subjected the purified Fc variants that were used in all assays in this study to SDS-PAGE and provided the Coomassie and silver staining images in **Extended Data Fig. 8** and related text in **P28, line 22-24** of the revised manuscript. And to address the reviewer's query on their purity, we quantified their purity based on their Coomassie staining (**Extended Data Fig. 8d**), which has a superior linear range. We have also retained the original Coomassie staining images in **Extended Data Fig. 1** as they serve to inform the relative expression of the Fc variants by the Expi293F prior to purification.

2. Ext Fig 3: The data on the disulphide-bonded derivatives is not fully convincing. How can authors be sure that the ds bridge is actually closed in the context of the Fc holo-molecule? The very slight shift in electrophoretic mobility could very well also come from the insertion of the cysteines as such. How about doing a confirmatory experiment with a corresponding isosteric double-serine variant?

Response: We agreed that the original data were not sufficiently convincing in demonstrating the formation of disulfide bonds in the ds derivatives and performed experiments to clarify this (**added as Extended Data Fig. 3**). Firstly, under reducing conditions, ds variants Fc(aMD4ds) showed only slightly lower electrophoretic mobilities than Fc(aMD4) because of the insertion of two cysteines in one chain (the theoretical molecular weights for Fc(aMD4) and Fc(aMD4ds) are 55,584 Da and 55,996 Da, respectively) (**added as Extended Data Fig. 3b, Reducing**). Despite their higher molecular weight, under non-reducing conditions, all Fc(aMD4ds) variants other than B1 clearly showed greater electrophoretic mobilities than Fc(aMD4), indicating a compact conformation due to the presence of an additional disulfide-linkage (**added as Extended Data Fig. 3b, Non-reducing**). Since we could not confirm the formation of the intended disulfide bond in Fc(aMD4ds)B1, the data derived from this mutant on Met activation and their description in the text were excluded in

the revised manuscript. Lastly, to provide more direct evidence, tryptic digests of Fc(aMD4ds)T1 and Fc(aMD4ds)B3 were analyzed by LC-MS/MS to detect peptides derived from disulfide-linked insertion sequences (added as Extended Data Figure 3c-e). Based on these results, we conclude that Fc(aMD4ds)T1/B2/B3 forms disulfide bonds at the insertion sequence. In view of these direct evidence of disulfide-linkage by LC-MS/MS, we did not conduct the comparison with a double-serine variant as the reviewer had helpfully suggested.

Results

3. Western blot in Fig. 1d could be improved; phospho-AKT etc. bands run funny (smiley shape).

Response: All western blotting related to original Fig. 1d has been repeated with improved band morphologies and providing the same observation (added as Fig. 1e and Supplementary Fig. 3).

4. How can it be explained that for aMD5, B1 was the best grafting site, whereas it is B3 for aMD4? Are there any conspicuous residues in the sequences that could explain this difference? I would suggest to run sequence comparisons (hydrophobicity, charges, bulkiness of residues etc.) that could give an insight, and perhaps include such bioinformatic data as a Suppl. Table.

Response: As aMD4 and aMD5 peptides have entirely different amino acid sequences, they could be considered distinct categories of Met dimerizers, especially given their different binding sites on the Met ectodomain (data not shown). As the reviewer recommended, we conducted structure prediction for the CH3 domain of Fc with aMD4 or aMD5 inserted in the B1 or B3 loop, using the AlphaFold2 software package (added as Supplementary Fig. 2). The results indicated the different spatial arrangements of the inserted peptides in each case. For aMD4, the pharmacophore sequence of the peptide is well exposed and accessible when inserted in B3, but may be structurally hindered when inserted in B1. Thus, the difference in the graft-site dependency of the two peptides is likely due to the steric effects imposed by their difference in the binding site on the Met ectodomain and their divergent spatial arrangements in the Fc structure. The above elaborations are added to the Result section of the revised manuscript (P5, lines 7-10).

5. The phospho-Met WB of the primary hepatocytes in Fig. panel 1d is not very convincing; please consider running this again or switch to a "better" anti-phospho-Ab.

Response: We repeated this experiment using larger quantities of hepatocyte lysate and antibody for immunoprecipitation. Similar results were obtained with improved band intensity (added as Fig. 1e and Supplementary Fig. 3).

6. The transcriptomics analysis performed on the human hepatocyte spheroids is very valuable! Any explanation, why in some cases the duplicates differ more than the different treatment regimens?

Response: At present, we do not know the reason for the variability between the transcriptomes of replicate hepatocyte spheroids between experiments. However, we note that similar variabilities could be observed between the experimental replicates in published studies (see figures below), suggesting that a degree of intrinsic variation exists between hepatocyte spheroids. This underscores the importance of biological replicates in transcriptomic analyses of this nature.

Can authors comment on the (few) differences observed between HGF treatment and the Fc variants? For example, *FGF19*, *SCL2A*.

Response: We re-evaluated the differentially expressed gene (DEG) of Fc(aMD4)B3-treated hepatocytes spheroids relative to HGF-treated hepatocytes spheroids and the resultant heatmap is shown in panel **a** in the figure below. As a result, we realized that only 37 DEG (*i.e.* 0.14 %) were of statistical significance with reproducibility between two experiments. As shown in panel **b**, most of these genes are similarly induced or repressed by the two treatments compared to non-treated spheroids, further emphasizing the similarity of gene expression changes between Fc(aMD4)B3-treatment and HGF-treatment. In general, the gene expression changes induced by Fc(aMD4)B3 are more pronounced than HGF, though the reason for this is currently unknown. Only *HNRNPR*, *TSTA3*, and *U2AF1L5* were oppositely altered by Fc(aMD4)B3-treatment and HGF-treatment with the reproducibility between two experiments, although the reason for this is also unknown. As noted in the reviewer's comments, *FGF19* and *SCL2A* expression were oppositely altered between Fc(aMD4)B3-treatment and HGF-treatment, but they were not contained in the 37

DEGs mentioned above. So, we have decided to remove the data in the revised Fig. 1j due to a lack of reproducibility between two experiments. We decided not to involve the below figures and related description/discussion in the text because they may cause confusion among readers by giving too much detail, which is not the main purpose of this paper.

a) Heatmap of differentially expressed genes of Fc(aMD4)B3-treated hepatocyte spheroids relative to HGF-treated hepatocyte spheroids at 24 h (E.1: experiment 1, E.2: experiment 2, False discovery rate < 0.05, Benjamini–Hochberg, two-sided).

b) Heatmap of gene expression of Fc(aMD4)B3-treated or HGF-treated hepatocyte spheroids relative to untreated hepatocyte spheroids at 24 h (E.1: experiment 1, E.2: experiment 2, False discovery rate < 0.05, Benjamini–Hochberg, two-sided).

7. Comment/appraisal: The Met-KO versus reconstituted CHO-Met approach is very elegant and offers strong evidence for binding.

Response: We appreciate this comment.

8. I am a bit puzzled by the data in Figure 2A and Extended Data Fig. 4. The mapping basically suggests that the agonistic Fc variants bind to a different site on Met than HGF. This would imply that the lasso-grafts behave as allosteric agonists... with all implied pros and cons. Could the authors comment on this?

Response: First, please note that our Fc-based dimerizers can activate Met on their own in the absence of HGF. So, they are not allosterically modulating the ligand function. There is no

evidence to support that activation of Met receptor by Fc(aMD4) is an allosteric effect. At present, we conclude that Met activation is due to the proximity between receptors induced by Fc(aMD4). Cytokine receptors and growth factor receptors are activated as a result of induction of proximity between receptors by agonist antibodies, antibody fragments, DNA aptamers, and dimeric cyclic peptides., *etc.* This point has been clarified in the revised version (P6, line 34 to P7, line 2).

9. Figure 2B: from the molecular weight markers that were co-electrophoresed, it is hard to actually claim that the highest band corresponds to a 2:1 complex. 2:2 might also be possible. But of course, in conjunction with the super-cool AFM experiments, the conclusion appears justified. Still, authors should consider re-doing the electrophoresis, e.g. using another gel (gradient gel, low percentage etc.).

Response: To better resolve the Met_{ECD}/Fc(aMD4) complex at high molecular weight, this experiment was performed again using low percentage gels (3-10% gels) and high molecular weight markers (added as Supplementary Figure 5a, b). Molecular weights analyzed by SDS-PAGE showed 2:1 and 1:1 complexes of Met_{ECD} and Fc(aMD4)B3. Together with HS-AFM analysis of the purified fraction of the highest band, we conclude that the highest band corresponds to a 2:1 complex.

10. It would be interesting, in Figure 4, to counterstain the brain sections for an astroglia (GFAP) and a microglia (Iba1) marker, because of potential applications for glioblastoma and astrocytoma.

Response: We appreciate the helpful suggestion relating to potential clinical applications. We performed counterstain for GFAP and Iba1 in sTfR(aMD4)-treated brain sections (added as Extended Data Fig. 7). sTfR(aMD4) diffused into the brain parenchyma and was detected in NeuN-positive neurons, but also in GFAP-positive astrocytes and Iba1-positive microglial cells. Related text is included in the text (P8, line 32-34). As suggested by the reviewers, these results indicate that our strategy can target a wide range of cell types in the brain parenchyma. Please note that sTfR(aMD4) diffuses into the mouse brain parenchyma independently of mouse Met receptor expression, since aMD4 does not interact with mouse Met.

What about the Met receptor? Is that expressed in brain?

Response: Met is expressed in the nervous system from pre-natal development to adult life, where it is involved in neuronal growth and survival (reviewed by Desole *et al.*, *Front. Cell Dev. Biol.*, 9, 683609, 2021, Ref. 34). We have confirmed that Met is expressed in neurons in mouse brain (added as Supplementary Fig. 7). HGF-induced Met activation exerts beneficial neuroprotective effects in preclinical models of cerebral ischemia, spinal cord injuries, and neurological pathologies,

such as Alzheimer's disease, amyotrophic lateral sclerosis, and multiple sclerosis (Ref. 34). Therefore, for brain diseases, systemic delivery of sTfR(aMD4) into the brain parenchyma could be a more realistic therapeutic option than HGF, which does not cross BBB. This statement was included in the text (P8, line 6-10).

Minor criticisms and comments

1. Discussion

Authors state "One major drawback of current BBB permeating antibodies is that their low brain-to-serum ratio and long half-life in circulation would often result in systemic concentrations that exceed the desired range for extended periods of time." Are they speculating about the BBB permeation of antibodies such as Aducanumab? If so, how could the current lasso-grafting approach come into play?

Response: We wanted to point out the potential danger of maintaining the high blood concentration of therapeutic antibodies targeted at central nervous system, if the antigen is also expressed in everywhere else. This may not be applicable to Aducanumab against A β aggregates in the brain, but highly relevant to the cases that the target molecule is functionally important in other tissues. The long residency of brain-targeting antibodies in systemic would cause unwanted neo-immunogenicity or side effects. With lasso-grafting, this could be avoided by grafting functional peptides directly into scaffold proteins with a short half-life, such as the anti-TfR Fab. We clarified this point in the Discussion (P18, line 9-15).

2. Abstract and general

The term "thioether"-macrocyclic peptide should be avoided or down-toned, as in the end not the initially screened-for thioether peptide is used but a linearized or ds variant.

Response: We agree with the reviewers' suggestions and amended to not use "thioether" except in one place in the Introduction (P3, line 29).

3. Introduction

When it comes to therapeutic targeting, the focus typically is on inflammatory or pro-cancerogenic cytokines such that the approach then is a "blocking" one. Exceptions next to HGF (which rather is classified as a growth factor) would be IL-10 and perhaps TGF β . Accordingly, there is a limited number of cytokine-based "agonist" approaches. Maybe, authors could expand their discussion a bit more and list and discuss potential other "agonistic" targets to further underscore the potential utility of the here presented lasso-grafting approach.

Response: We are grateful to the reviewer's advice aimed to help us highlight the potential utility

of the lasso-grafting approach. Indeed, a major attraction of this approach is the relative ease in engineering agonistic macrocyclic molecule into a well-understood scaffold. With this in mind, we have elaborated where agonistic therapeutics would be desirable in the opening paragraph of the introduction (P3, line 1-5):

The clinical utilization of cytokines and growth factors as therapeutics has been approved by FDA^{1,2} and their potential application in emerging areas, such as neurogenesis and brain repair^{3,4} is a topic of intense research. However, their inherent structural properties render them challenging to engineer for improved physicochemical stability and pharmacokinetics, in particular better half-life or BBB penetrance.

Although there are many examples where agonist therapeutics are being actively researched (e.g., NGF, BDNF, PDGF, EGF, IGF in Ref. 3, TrkB in Ref. 4, Wnt in Ref. 11, IL-2, IL-10, IFN in Ref. 12, etc.) for the treatment for various diseases, we have refrained from listing specific examples in the interest of space. Rather, we have highlighted the Achilles' heel of such approaches, namely the natural short half-life of cytokines and growth factors in circulation and their inability to cross the BBB. This was followed by emphasizing the potential of lasso-grafting in agonists development, as the reviewer kindly recommended, taking into consideration promising recent progresses (P3, line 11-15):

To overcome the inherent structural limitations of cytokines and growth factors, there has been progress in the development of surrogate agonists that are structurally unrelated to native ligands^{11,12}. Despite these recent progresses, there remains a large unmet need for more robust and versatile methods to design protein therapeutics with desired physicochemical stability and pharmacokinetics.

As with above, we have refrained from describing these specific recent examples of vastly different approaches to keep the Introduction concise and focused. As part of these elaboration of lasso-grafting's potential in agonist designs, the following articles were cited and added in the reference list:

1. Ray, A., Gulati, S.G.K., & Joshi, N.R.J. Cytokines and their role in health and disease: A brief overview. *MOJ Immunol.* **4**, 00121. <https://doi.org/10.1540> (2016).
2. Silva, A.C., & Sousa Lobo, J.M. Cytokines and Growth Factors. *Adv Biochem Eng Biotechnol.* **171**, 87–113. https://doi.org/10.1007/10_2019_105 (2020).
3. Oliveira, S.L., et al. Functions of Neurotrophins and Growth Factors. *Cytometry A.* **83**, 76–89. <https://doi.org/10.1002/cyto.a.22161> (2013).
4. Wang, S. et al. Therapeutic potential of a TrkB agonistic antibody for Alzheimer's disease. *Theranostics.* **10**, 6854–6874. [https://doi: 10.7150/thno.44165](https://doi.org/10.7150/thno.44165) (2020).

11. Janda, C.Y. *et al.* Surrogate Wnt agonists that phenocopy canonical Wnt and β -catenin signalling. *Nature*. **545**, 234–237. <https://doi.org/10.1038/nature22306> (2017).
12. Yen, M. *et al.* Facile discovery of surrogate cytokine agonists. *Cell*. **185**, 1414 – 1430.e19 <https://doi.org/10.1016/j.cell.2022.02.025> (2022).

4. Results

Although published previously, the actual “grafting” principle should be described better. This reviewer has a biochemistry background, but I had to go back to virtually all previous publications in NCOMMS, Nat Chem Biol, and Angewandte to be able to fully understand how this works.

Response: We thank the reviewer for highlighting this important oversight and appreciate the reviewer’s effort to understand our previous works. In the revised manuscript, we have expanded our introduction of lasso-grafting to include a brief description of its biochemistry, summarizing our previous works (P3, line 19-21, P3, line 27 to P4, line 2).

To the end, how important is the thioether bond present in the initially identified Met-binders?

Response: The thioether bond is essential to cyclize the initially identified Met-binders. In general, macrocyclic peptides display greater affinities for targets than the linear peptides due to their constrained, cyclic structures. The linearized versions of Met-binders (aMD4, aML5, aMD5) have 70 to 300-fold less affinities (Ito *et al.*, *Nat Commun.*, 6, 6373, 2015, Ref. 39, see figure below). However, the thioether bond moiety itself is not important because it is generally pointing away from the target proteins and does not contribute to the binding. Therefore, lasso-grafting the internal sequence without thioether or disulfide knot is generally sufficient to grant target-binding ability because they look like “cyclic” after the loop presentation (for example, see predicted structures added as Supplementary Fig. 2). Related text was included in P3, line 19-21, P3, line 27 to P4, line 2, and P5, line 20-22.

Modified from Ito, *et al.*, *Nat Commun.* 6, 6373. (2015)

	Macrocycle	Sequence	k_a ($10^6 \text{ M}^{-1} \text{ s}^{-1}$)	k_d (10^{-2} s^{-1})	K_D (nM)
Macrocycles	aML5	Ac- ^L -YISWNEFNSPNWRFITCG-NH ₂	0.43	0.81	19
	aMD4	Ac- ^D -YRQFNRRTHEVWNLDCG-NH ₂	1.4	0.34	2.4
	aMD5	Ac- ^D -YWYYAWDQTYKAFPCG-NH ₂	4.8	1.1	2.3
	aMsD4	Ac- ^D -YERVNHLFRNQTWDRCG-NH ₂		No binding	
Linearized	aML5-Lin	Ac- ^L -YISWNEFNSPNWRFITAG-NH ₂	0.013	1.7	1300
	aMD4-Lin	Ac- ^D -YRQFNRRTHEVWNLDCG-NH ₂	0.55	4.0	750
	aMD5-Lin	Ac- ^D -YWYYAWDQTYKAFPCG-NH ₂	0.012	0.23	190

Along the same lines, what changes may be induced by the spacer glycine-residues?

Response: Linker is not always required, but depending on the peptide sequence and binding site, the presence of a linker may improve accessibility, so a glycine linker of 1 to 2 amino acids is included on each side as a default. In the present case, the peptide shows sufficient affinity and agonist activity, so no optimization was performed. However, if the affinity is unsatisfactory, randomizing and reselecting lengths and amino acid compositions of linker can dramatically improve affinity in some cases.

5. typo "yeilding" in methods.

Response: This was corrected (P28, line 25).

Reviewer #3 (Report for the authors (Required)):

In this manuscript, Sakai and colleagues describe thioether-macrocyclic peptide pharmacophores grafted into Fc-receptor scaffold to generate therapeutic proteins with pharmacological activity in vivo. This work builds on a platform recently developed by this group for selecting and stabilizing macrocyclic peptides by grafting into a protein scaffold (lasso-grafting) (Mihara et al. Nature Comms 2021, ref 21) In this work, lasso-grafted FcR with Met receptor ligands are optimized further for high-affinity binding to c-Met. In addition to showing activation of the Met receptor in cells stimulated ex-vivo, the authors study the in vivo properties of these lasso-grafted molecules in mice engrafted with human hepatocytes. As expected from the Fc scaffold, the grafts displayed extended half-lives of ~50 hours compared to the anti-transferrin receptor antibody conferred significant brain penetration, particularly with a single-arm TfR configuration

Overall, this paper represents an advance in understanding and translatability of this potential therapeutic platform over the group's previous work, with the first demonstration of in vivo PD with this type of graft and use of the known ability of anti-transferrin receptor antibody to endow brain penetrance. However, there are a number of specific issues that should be addressed before publication.

Response: We are grateful for the reviewer's time and effort for the evaluation of our work and the insightful feedback aimed to strengthen our manuscript. Guided the reviewer's suggestions and we have conducted the following experiments to address them.

Major

1. Figure 1: The authors claim identical signaling for the lasso grafts but only show a single time point. Other time points should be studied for pERK or pMet also at lower doses given the slower on rate if not done in their previous work.

Response: In response to the reviewer's concern, we examined the induction of Met, Akt, and Erk phosphorylation by HGF or Fc(aMD4)B3 at maximal or low doses at multiple time points (0, 3, 10, 20, 30, 90, and 180 min) (added as Extended Data Fig. 4b, related text in P5, line 27-30). Overall, the induction levels as well as the induction kinetics of Met, Akt, and Erk1/2 phosphorylation were highly comparable between HGF and Fc(aMD4)B3, except that low-dose Fc(aMD4)B3 induced Akt phosphorylation slightly faster than low-dose HGF.

As the Lasso graft does not compete with HGF – could that activate different phosphorylation sites

on cMet? Have they done any mass spec to probe this in an objective way?

Response: It has been well established by others that in addition to tyrosine 1234/1235, Met tyrosine 1003 and 1349 are important for signal activation. Therefore, we analyzed the phosphorylation of these tyrosine residues induced by HGF and Fc(aMD4)B3 by Western blotting (added as Extended Data Fig. 4a, related text in P5, line 27-30). HGF and Fc(aMD4)B3 similarly induced phosphorylation of these tyrosine residues. Together with the time course of pMet, pAkt, and pErk described above, these results are consistent with the notion that Fc(aMD4)B3 activates Met through a manner highly comparable to HGF.

2. Figure 2: Met dimerization induced by the aMD4 macrocycle graft that is shown should be compared to HGF induced dimerization. The relative ability of the lasso grafts vs HGF to induce receptor dimerization is not clear from data presented.

Response: The reviewer has asked for a comparison of the relative ability of lasso-grafted Fc and HGF to induce receptor dimerization. However, our Met dimerization assay is not the best method to make such a comparison as the Met dimerization efficiency measured by this assay was much lower than that measured in the cellular context examined in our previous study (Ito *et al.*, *Nat Commun.*, 6, 6373, 2015, Ref. 39). In that study, the Met dimerization efficiency between HGF and dimer-aMD4 peptides were compared within the cellular context (provided below for the reviewer's reference). The discrepancy between the two assays is likely due to: 1) the use of a truncated Met_{ECD} instead of native full-length Met; 2) Met_{ECD} proteins are dispersed in a random configuration in solution (in cells, Met is maintained in a consistent configuration by the plasma membrane); 3) low concentration of protein (20-60 nM) results in insufficient cross-linking efficiency by bis(sulfosuccinimidyl)suberate (BS³). As such, we have reserved the Met dimerization assay for the expressed purpose of elucidating the structure of the Fc(aMD4)-Met dimer complex in Fig. 2 and not for comparisons of Met dimerization efficiency. Instead, as the phospho-Met (Tyr1234/1235) in cells most closely reflects Met dimerization efficiency under physiological conditions, we have measured the relative amount of phospho-Met (Tyr1234/1235) induced by the Fc variants compared to the maximum level of phospho-Met (Tyr1234/1235) induced by HGF (Fig. 1b and c). To better address the reviewer's request, we have further quantified the levels of phospho-Met (Tyr1234/1235) induced by various concentrations of HGF, Fc(aMD4)B3, and Fc(aMD4ds)B3 and added these data as Figure 1d.

3. Figure 4: Although one of the more novel aspects of the paper, some of the brain penetration data are not clear and fall short of the rigor of the other experiments:

a. Why are different mg/kg doses used for the different anti-TfR fusions (line 312)?

Response: An equimolar amount/kg was used for better comparison of Fc(aMD4), sTfR(aMD4), and dTfR(aMD4). Since the molecular weights are different, we used different amounts of each Fc variants to maintain an equimolar dose per animal. We clarified this point in the revised text (P8, line 23 in the Result; P16, line 9 in the Fig. 4 legend; P35, line 25 in the Method).

b. In addition to the brain/serum concentration ratio, a brain/body ratio should be calculated to understand the biodistribution more globally. This value will help clarify what percentage of the molecule enters the brain relative to the dose administered in a manner that allows comparison with other technologies. The typical levels reported for other technologies are 1-2% entering the brain, which is in line with the weight ratio of the brain to body.

Response: We appreciate this helpful suggestion on better data presentation. The % injected dose/gram brain was calculated for Fc(aMD4), sTfR(aMD4), and dTfR(aMD4) (added as Fig. 4g, related text in P8, line 25-26). The value $0.96\% \pm 0.07\%$ for sTfR(aMD4) is close to the reviewer's estimates and the reported values (0.4 to 0.6%) for low-affinity TfR antibodies administered at therapeutic doses as shown below (Yu *et al.*, *Sci Trans Med.* 3, 84ra44, 2011, Ref. 9, see figure below).

c. Please explain the rationale for selecting the 24 hour time point for the brain experiments.

Response: The 24-hour time point was selected based on previous reports by Yu *et al.* (*Sci Trans Med.* 3, 84ra44, 2011, Ref. 9) and Couch *et al.* (*Sci Trans Med.* 5, 183ra57, 2013) that therapeutic doses of anti-TfR Abs peaked in the mouse brain after approximately 24 hours (see figures below). We also referred to a paper suggested by the reviewer (Chang *et al.*, *MAbs*, e1874121, 2021, added as Ref. 49) that examined a detailed PK profile (time course and brain region) of anti-TfR antibodies with various affinities for TfR in the rat brain (see figures below). According to this paper, anti-TfR antibodies peaked in the rat brain within a few hours and persisted for 24 hours. Hence, considering these previous reports, we believe the 24-hour time point is reasonable. That said, further evaluation of detailed PK profiles, including time course, will be needed for in future therapeutic evaluation of sTfR(aMD4) in preclinical models, which we noted in the revised manuscript (P9, line 3-5).

Therapeutic doses of anti-TfR Ab (20 mg/kg) and anti-TfR/BACE1 specific Abs (50 mg/kg) peak in the mouse brain around 24 hours. Yu, *et al.*, *Sci Transl Med.* 3, 84ra44. (2011) & Couch, *et al.*, *Sci Transl Med.* 5, 183ra57. (2013)

Therapeutic doses of Anti-TfR Ab variants (10 mg/kg) peak in the rat brain in hours to 24 hours.

K_D values to TfR: OX26-5 = 5 nM, OX26-76 = 76 nM, OX26-108 = 108 nM, OX26-174 = 174 nM.

Chang, *et al.*, *MABS* 13, e1874121. (2021)

d. *The biological activity of the Met agonists in the brain could not be tested in vivo as the target is not in the CNS and the macrocycles are not cross-reactive to mouse.*

Response: As noted by the reviewer, the high species specificity of our macrocyclic peptides for human Met precludes such experiments at this time. However, in the future, we would like to report such data after re-selecting our macrocyclic peptides for mouse Met. Although we have not clearly presented in the original text, Met is expressed in the nervous system, where it is involved in neuronal growth and survival (reviewed by Desole *et al.*, *Front. Cell Dev. Biol.*, 9, 683609, 2021, Ref. 34). In the revised manuscript, we have confirmed that Met is expressed in neurons in mouse brain (added as Supplementary Fig. 7). We wish to highlight that HGF-induced Met activation exerts beneficial neuroprotective effects in preclinical models of cerebral ischemia, spinal cord injuries, and neurological pathologies, such as Alzheimer's disease, amyotrophic lateral sclerosis, and multiple sclerosis (Ref. 34). However, as HGF cannot cross the BBB, systemic delivery of sTfR(aMD4) into the brain parenchyma could represent a more viable therapeutic option to HGF for the treatment of brain associated pathologies. We have added this point in the revised text (P8, line 6-10).

For example, in their previous paper the authors develop a binder to PlxnB1 which is a CNS-expressed gene.

Response: Thank you for this suggestion. We aim to report such data in the future. However, the activation of PlexinB1, a GTPase-activating protein, cannot be easily detected by immunohistochemical methods. Therefore, PlexinB1 activity in the brain can only be evaluated using biological or therapeutic readouts. The limited time available for resubmission does not allow for such experiments. We would appreciate the reviewer's understanding that this is also beyond the scope of our manuscript.

4. Discussion: The authors should compare the merits of this novel format vs antibody-based biologics such as bispecific anti-TfR/anti-cytokine receptor agonists both for therapeutic value and in terms of potential immunogenicity.

Response: As the reviewer points out, bispecific anti-TfR antibodies have been explored to endow BBB-penetrating ability to various anti-receptor antibodies, mainly for blocking purpose. We found only one anti-TfR/anti-cytokine receptor (TrkB) agonist. (Clarke *et al.*, *bioRxiv*, doi:<https://doi.org/10.1101/2020.03.12.987313>; **added to the reference list**). Although many different bispecific formats have been reported, they generally involve fusion of two antigen binding domains via highly sophisticated protein engineering methods. This would in turn result in bispecific antibodies that are complex and artificial often failing to meet the manufacturing yield and stability required for drug development. In contrast, our lasso-graft method changes the original IgG structure minimally by merely adding an appendage of ~15-residue at the tip of exposed loop, which rarely affects the expression yield and stability of the parental protein (Extended Data Fig. 1). The minimum modification and absence of large introduced protein domain from non-human origin (such as shark antibody, Clarke *et al.*) also contribute to the lower risk of immunogenicity. In the revised manuscript, these benefits are outlined in the Discussion section (**P18, line 5-9, 16-23**). Also noted, is the applicability of lasso-grafting to a variety of protein scaffolds that could allow for better brain-targeted therapeutics with less systemic distribution, among other benefits (**P18, line 9-15**).

Is there any data on immunogenicity of the lasso grafts?

Response: Immunoinformatic prediction and *ex vivo* T cell assays were performed to predict the potential antigenicity of lasso-grafting Fc (See below figure for reviewers only, the analytical work was outsourced to EpiVax, Inc.). In summary, both assays showed the minimal potential of Fc(aMD4)B3 and Fc(aMD5)B1 for immunogenicity as compared to the control Fc scaffold. The

immunogenicity of lasso-grafted Fc is likely to depend on both peptide sequence and insertion site and should be carefully evaluated in each individual case during drug development process. This statement was included in the revised Discussion (P18, line 21-23). This information is provided exclusively for the review of this manuscript, as the data cannot be included in the revised manuscript due to their proprietary nature.

Minor

2. In Figure 1, in addition to the heat map shown, a dot plot should be shown for gene induction in response to HGF vs Fc(aMD4) to better appreciate the correlation between the gene expression changes induced by the two ligands.

Response: We appreciate this thoughtful comment. As the reviewer requested, we generated scatter plots showing the correlation between the gene expression changes induced by HGF and Fc(aMD4)B3 as shown below. One of these is added as Fig. 1h in the revised manuscript and is described on P5, line 34-35.

2. The authors should cite <https://www.ncbi.nlm.nih.gov/pmc/articles/PMC7849817/> and discuss their transferrin affinity in that context (they used an older paper).

Response: Thank you for this information. This article has been added to the text (P8, line 28-29) and to the reference list (Ref. 49).

Rebuttal 2

Point-By-Point Response to the Reviewers' Comments

We deeply appreciate the time and effort each of the reviewers have dedicated to carefully evaluating our manuscript and providing insightful feedbacks aimed to strengthen the quality of our paper. In the following paragraphs, we respond to the comments made by each reviewer. Our responses are in black while the reviewers' comments are in blue and italicized.

Reviewer #2 (Report for the authors (Required)):

Authors offer an extensively and carefully revised manuscript and, overall, have satisfactorily responded to my critique.

Regarding my second point (How can authors be sure that the ds bridge is actually closed in the context of the Fc holo-molecule?), I am fine with their approach of running reduced versus non-reduced gels plus LC-MS digests. The results convincingly show that the disulfide has formed for T1, B2, and B3. And I agree that with these data, the analysis of a double serine mutant wouldn't be necessary.

Response: We would like to thank the reviewer again for her/his efforts in evaluating our

paper and providing us with insightful feedback to improve the quality of our paper.

The indicated molecular weights in new Extended Figure 3b are, however, a bit confusing for the reader. I assume the non-reducing panel refers to an Fc dimer, while the reducing one to a monomer. I would recommend to somehow indicate and clarify this in the figure. Alternatively, full size gels from 10 100 kD or so could be shown for both conditions.

Response: Following this suggestion, we have indicated the term " dimer" for non-reducing SDS-PAGE and "monomer" for reducing SDS-PAGE of Extended Data Fig. 3b, as shown in the figure below.

Thank you for clarifying as to my point #8. I actually didn't think of true "allostery" with the dimerizers binding to Met "in addition" to the endogenous agonist, but actually had used the word "allosteric" just to indicate binding at a different site compared to HGF. I think the authors explanations have clarified my point, although, I have seen this right, Reviewer # 3

had a similar issue.

Response: Thank you for explaining what you meant by allostery. This is consistent with what we understood you intended. We are also pleased that the revised text fully clarifies the questions you and reviewer #3 raised.

Reviewer #3 (Report for the authors (Required)):

I have reviewed the authors' point-by-point response and their response fully addresses the concerns. Thank you for carefully addressing them, and my apologies that this re-review took some time.

Response: We would like to thank the reviewer again for her/his efforts in evaluating our paper and providing us with insightful feedback to improve the quality of our paper.

Reviewer #4 (Report for the authors (Required)):

This is an elegant study that marries the advantages of a Fc domain with peptides that bind

to an epitope. The authors have addressed the comments that were raised by the reviewers. While this builds on the previous studies from the same group on this platform, the fact that here they show Met agonism with the platform is extremely relevant and could have high impact. This is because MET agonists are needed as therapeutics. Data presented is strong from a biochemical and signaling perspective. The use of humanized liver implants is also elegant.

Response: We are very grateful to the reviewers for their appreciation for the importance of this paper and for their supportive comments.

While the authors show brain delivery, this is a distraction from what could be a potential translation of the product. HGF cannot be used as a therapeutic as it degrades in circulation (which could also explain the PK data in this paper) and its heparin binding limits its access into tissues. A potential therapeutic use of this agonist vs HGF could have been in NASH models or in models of liver failure. The prolonged circulation time of the agonist, and potentially absence of heparin binding (need to test for that) could have given a significant therapeutic advantage over HGF. Efficacy in a disease model would significantly increase the impact.

Response: We agree that the therapeutic efficacy of designer agonists is needed to fully

justify their use as therapeutic agents. Following this suggestion, we have added a brief discussion of the suitability of the agonist for potentially treating liver disease and necessity to evaluate therapeutic efficacy of designer agonists in animal models in the discussion (p10-11, line 319-323). Regarding the heparin-binding property, the native ligand HGF binds strongly to heparin, whereas the lasso-grafted Fc mutant does not (Please see the figure below). This data is useful for future animal disease models, but is not included in this article as it is not essential to support the conclusions of this manuscript.